# Mode of delivery modulates the intestinal microbiota and impacts the response to vaccination

Emma M. de Koff [1,2,6], Debbie van Baarle[3,7], Marlies A. van Houten[1,4], Marta Reyman [2,8], Guy A. M. Berbers[3], Femke van den Ham[3], Mei Ling J. N. Chu[2,3], Elisabeth A. M. Sanders [2,3], Debby Bogaert [2,3,5,9] & Susana Fuentes [3,9]

The gut microbiota in early life, when critical immune maturation takes place, may influence the immunogenicity of childhood vaccinations. Here we assess the association between mode of delivery, gut microbiota development in the first year of life, and mucosal antigen-specific antibody responses against pneumococcal vaccination in 101 infants at age 12 months and against meningococcal vaccination in 66 infants at age 18 months. Birth by vaginal delivery is associated with higher antibody responses against both vaccines. Relative abundances of vaginal birth-associated *Bifidobacterium* and *Escherichia coli* in the first weeks of life are positively associated with anti-pneumococcal antibody responses, and relative abundance of *E. coli* in the same period is also positively associated with anti-meningococcal antibody responses. In this study, we show that mode of delivery-induced microbiota profiles of the gut are associated with subsequent antibody responses to routine childhood vaccines.

Vaccination in early childhood is estimated to save millions of lives each year[1]. Vaccine-induced protection is mediated through a combination of innate, humoral and cellular immunity, and is often quantified by measuring antigen-specific antibody titers[2]. Large interindividual variation in antibody responses to vaccines administered in early life may limit vaccine effectiveness, leaving some fully vaccinated infants unprotected against serious infectious diseases[3]. Factors that influence vaccine responses include, among others, genetics, sex, perinatal characteristics like gestational age, birth weight, maternal antibodies, and feeding type, but also more general

factors like geographical region (reviewed in[4]). Recent research has shown that the gut microbiota, i.e. the sum of all microorganisms residing in the human intestinal tract, also play a role in immune responses to vaccination[5–11]. This offers a potentially modifiable target to improve immunogenicity of childhood vaccines.

The gut microbiome is seeded at birth and rapidly develops over the first months of life under the influence of mode of delivery, breastfeeding, antibiotic administration and nutrition[12–15]. Timely exposure to specific microbes within the critical window of opportunity in early infancy shapes the immune system[16–18], including the B cell

[1]Spaarne Academy, Spaarne Gasthuis, Hoofddorp and Haarlem, Netherlands. [2]Department of Paediatric Immunology and Infectious Diseases, Wilhelmina Children's Hospital and University Medical Centre Utrecht, Utrecht, Netherlands. [3]Centre for Infectious Disease Control, National Institute for Public Health and the Environment, Bilthoven, Netherlands. [4]Department of Paediatrics, Spaarne Gasthuis, Hoofddorp and Haarlem, Netherlands. [5]Medical Research Council and University of Edinburgh Centre for Inflammation Research, Queen's Medical Research Institute, University of Edinburgh, Edinburgh, UK. [6]Present address: Department of Medical Microbiology and Infection prevention, Amsterdam University Medical Centre, Amsterdam, Netherlands. [7]Present address: Department of Medical Microbiology and Infection prevention, Virology and Immunology research Group, University Medical Centre Groningen, Groningen, Netherlands. [8]Present address: Department of Dermatology, Erasmus University Medical Centre, Rotterdam, Netherlands. [9]These authors jointly supervised this work: Debby Bogaert, Susana Fuentes. ✉e-mail: d.bogaert@ed.ac.uk

and immunoglobulin repertoire[19,20]. Microbial imprinting on the immune system in early life may in turn explain part of the variation in vaccine responses. In support of this hypothesis, it has been shown that antibiotic-induced microbial perturbances in an infant mouse model led to impaired antigen-specific immunoglobulin G (IgG) responses against five common childhood vaccines including the meningococcal group C (MenC) conjugate vaccine and the 13-valent pneumococcal conjugate vaccine (PCV-13)[21]. Microbiota perturbance due to antibiotic exposure also resulted in impaired immune responses to seasonal influenza vaccination in healthy adults without pre-existing immunity, suggesting that primary responses are more sensitive to microbiota changes than recall responses[7]. In human infants, the composition of the microbial community pre-vaccination has been correlated with systemic immune responses to oral rotavirus vaccine, oral poliovirus vaccine, *Bacillus* Calmette-Guérin, hepatitis B, and tetanus vaccines[5,6,10,11,22,23]. However, the temporal relationship between (1) early life exposures, (2) gut microbiota composition, and (3) subsequent childhood vaccine responses has not yet been studied.

Here, we demonstrate in a healthy birth cohort that mode of delivery-induced differences in microbial colonization patterns in the gut in early life are associated with antigen-specific IgG responses to the 10-valent PCV (PCV-10) and the MenC conjugate vaccine in saliva. For these vaccines, mucosal IgG has been shown to confer vaccine-induced protection against infection[24]. These findings are key for the design of intervention strategies that modulate the gut microbiota to enhance vaccine immunogenicity in infants.

## Results

We investigated associations between early life exposures, gut microbiota development in the first year of life and its effect on vaccine responses later in life in a birth cohort of 120 healthy, term born infants[25]. Follow-up of the infants and sample inclusion for gut microbiota characterization by 16S rRNA gene sequencing and salivary antigen-specific IgG measurement by multiplex immunoassay are shown in Supplementary Fig. 1. Basic, lifestyle and environmental characteristics were previously published[26], and are briefly summarized in Table 1. Infants received routine vaccinations according to the Dutch National Immunization Program (NIP). Serotype-specific anti-pneumococcal IgG concentrations were measured in routinely collected saliva of 101 infants at the age of 12 months (median 28 days [IQR 21–33] after the PCV-10 booster dose). Anti-MenC IgG concentrations were measured in routinely collected saliva of 66 infants at the age of 18 months (median 116 days [IQR 105–120] after MenC vaccination). Geometric mean concentrations (GMC) of IgG concentrations against the different pneumococcal vaccine serotypes ranged from 7.33 ng/ml (95% CI 5.75–9.33 ng/ml) for serotype 23F to 27.30 ng/ml (95% CI 22.14–33.67) for serotype 19F. The anti-MenC IgG GMC was 10.64 ng/ml (95% CI 8.64-13.11 ng/ml; Fig. 1a). IgG concentrations against the 10 pneumococcal vaccine serotypes strongly correlated with each other (Pearson's $\rho$ 0.56-0.86, adjusted $p < 0.001$ for all pairwise correlations), and not with anti-MenC IgG antibodies (Pearson's $\rho$ 0.12–0.31, adjusted $p > 0.397$ for all pairwise correlations; Fig. 1b). As serotype-specific anti-pneumococcal IgG concentrations were strongly correlated, we focused our analyses on serotype 6B, which shows relatively weak antigenic properties, and is commonly found during (severe) pneumococcal disease[27]. Significant findings were validated for the other serotypes.

### Mode of delivery was associated with vaccine responses

We first investigated whether early life host characteristics previously associated with differences in gut microbiome development and/or vaccine immunogenicity, were related to anti-Ps6B and anti-MenC IgG responses. Mode of delivery, feeding type, sex, antibiotics use in the first three months of life, and pets in the household were related to vaccine responses against one or more serotypes in univariate analysis, while having older siblings, the number of antibiotic courses, and daycare

**Table 1 | Cohort description**

| | PCV-10 response | MenC response |
|---|---|---|
| *n* | 101 | 66 |
| Sex, female (%) | 54 (53.5) | 35 (53.0) |
| Perinatal characteristics | | |
| Mode of delivery, vaginal (%) | 58 (57.4) | 42 (63.6) |
| Antibiotics during birth (%) | 2 (2.0) | 1 (1.5) |
| Exclusive formula feeding (%) | 17 (16.8) | 10 (15.2) |
| Breastfeeding, days (median [IQR]) | 55.0 [3.0, 248.0] | 114.0 [3.0, 289.8] |
| Environmental characteristics | | |
| Presence of siblings (%) | 68 (67.3) | 46 (69.7) |
| Number of siblings (median [IQR]) | 1.0 [0.0, 1.0] | 1.0 [0.0, 0.1] |
| Pets in the household (%) | 46 (45.5) | 36 (54.5) |
| Antibiotic treatment | | |
| Antibiotics in the first 3 months (%) | 13 (12.9) | 4 (6.1) |
| Antibiotic courses[a] (median [IQR]) | 0.0 [0.0, 1.0] | 0.0 [0.0, 1.0] |

Participant characteristics are summarized for all infants who had anti-pneumococcal immunoglobulin G (IgG) responses available (n = 101; left column), and the subset of infants who had anti-meningococcal type C (MenC) IgG responses available (n = 66; right column). [a]The number of antibiotic courses is given up to the time that vaccine responses were measured, so up to 12 months in the left column and up to 18 months in the right column. Source data are provided in the Source Data file.

attendance were not. These variables were included in multivariable linear models, including an interaction term between mode of delivery and feeding type due to the interdependency of these variables. Vaginal delivery (in contrast to cesarean (C-)section birth) was independently associated with higher anti-Ps6B IgG concentrations ($\beta = 0.51$ [95% CI 0.043–0.97], $p = 0.033$; Fig. 2a). However, we also observed a negative interaction between vaginal delivery and exclusive formula feeding on anti-Ps6B responses ($\beta = -1.32$ [95% CI −2.43 to −0.21], $p = 0.021$), suggesting that the positive effect of vaginal birth was diminished by subsequent formula feeding. Similar associations were found for IgG responses to most of the other pneumococcal vaccine serotypes (Supplementary Table 1). Stratified analyses confirmed that, within the vaginally delivered group, the anti-Ps6B IgG GMC of breastfed infants ($n = 51$) was 3.5-fold higher compared to formula fed infants ($n = 7$; adjusted $p = 0.070$); similarly, within the breastfed group, the anti-Ps6B IgG GMC of vaginally delivered infants ($n = 51$) was twofold higher compared to C-section born infants ($n = 33$), although this difference was not significant (adjusted $p = 0.51$). Anti-Ps6B IgG concentrations did not differ between feeding types within the C-section born group (Fig. 2b). Likewise, for MenC, vaginal delivery was also associated with higher IgG concentrations compared to C-section delivery ($\beta = 0.42$ [95% CI 0.016–0.83], $p = 0.042$), which was independent of feeding type (Fig. 2a). In a stratified analysis, vaginally delivered infants ($n = 42$) showed a 1.7-fold higher anti-MenC IgG GMC compared to C-section delivered infants ($n = 24$; $p = 0.002$; Fig. 2b). Mode of delivery and feeding type were thus the only early life factors significantly associated with IgG responses against Ps6B and MenC, while sex, antibiotic use, and having pets were not. We concluded that mode of delivery and feeding type are likely microbiome modulators from birth onward[26], and therefore considered them as such for our downstream analysis.

### Gut microbial community composition at one week of age was associated with vaccine responses

We then studied whether gut microbiota development in the first year of life was associated with anti-Ps6B and anti-MenC IgG responses.

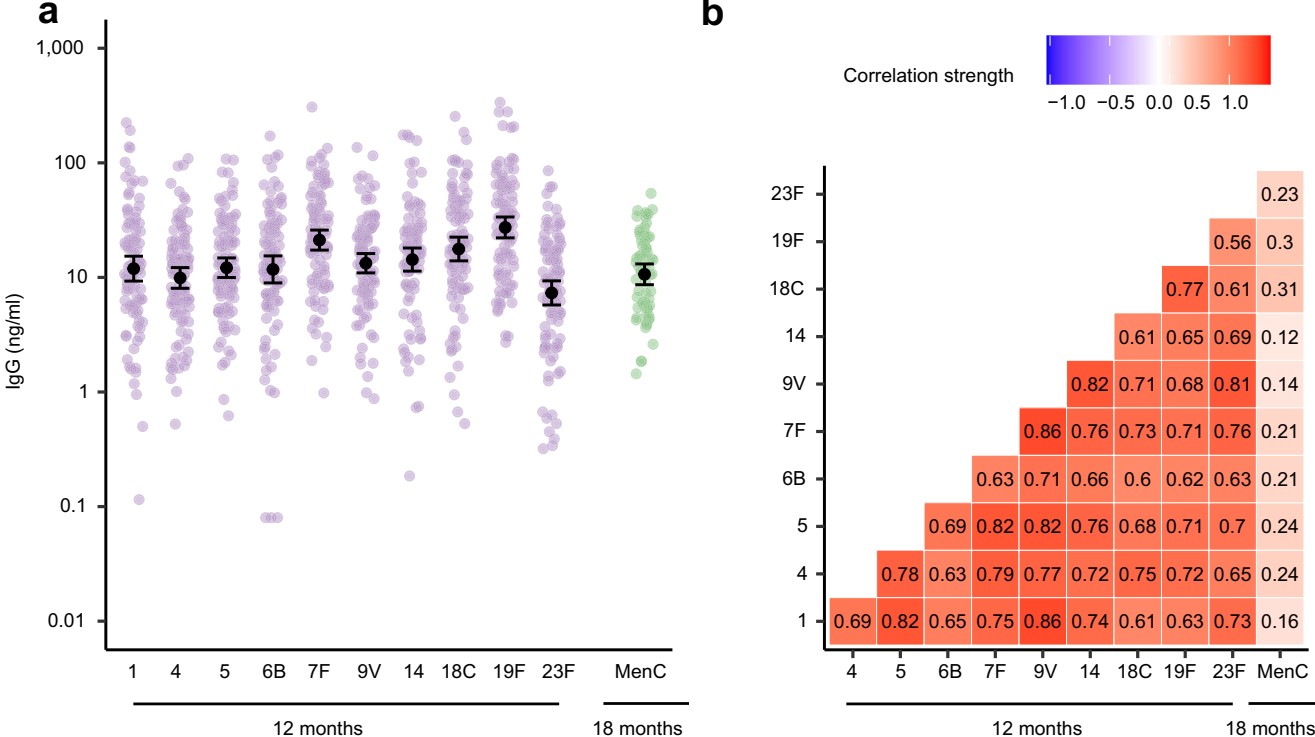

**Fig. 1 | Anti-pneumococcal and anti-MenC IgG concentrations following vaccination. a** Immunoglobulin G (IgG) concentrations against 10 pneumococcal vaccine serotypes (1, 4, 5, 6B, 7F, 9V, 14, 18C, 19F, and 23F; $n = 101$) and meningococcus type C (MenC; $n = 66$) following vaccination. Black dots and error bars represent geometric mean concentrations with 95% confidence intervals (CI).

**b** Correlation plot of IgG concentrations against the 10 pneumococcal vaccine serotypes and against MenC following vaccination. Numbers indicate the correlation strength, which was evaluated using Pearson's correlation coefficients. Source data are provided in the Source Data file.

Overall, 1052 out of 1156 fecal samples (91.0%) passed quality control for 16S rRNA gene-based sequencing, and were included in further analyses (Supplementary Fig. 1). We have previously shown in this cohort that the gut microbiota composition of infants born by C-section was significantly different compared to vaginally delivered infants, with lower relative abundance of *Bifidobacterium* and *Escherichia coli*, and enrichment of *Enterococcus faecium* and *Klebsiella*, from birth persisting up to the age of two months[26]. From the age of two months onward, the gut microbiota composition remained comparable between mode of delivery groups.

We first studied associations between the alpha diversity measures, including Shannon diversity and the observed number of species, at each time point and vaccine responses. No association was found between alpha diversity and anti-Ps6B or anti-MenC IgG concentrations at any time point, with the exception of an inverse correlation between the observed number of species at the age of two months and anti-Ps6B IgG concentrations ($\beta = -0.029$ [95% CI $-0.049$ to $-0.0087$], adjusted $p = 0.082$). This association was not observed for the other pneumococcal vaccine serotypes.

We compared the overall microbial community composition between infants with above and below median anti-pneumococcal and anti-meningococcal IgG responses using permutational analysis of variance (PERMANOVA) on the Bray-Curtis dissimilarity matrix per timepoint, and found no significant differences. As a measure of gut microbiota stability, we calculated the Bray-Curtis similarity (1-Bray-Curtis dissimilarity) between consecutive timepoints within individuals. Microbiota stability between day one and week one, and between week one and week two correlated with higher anti-Ps6B IgG concentrations (day 1-week 1: $\beta = 1.66$ [95% CI 0.44–2.88], adjusted $p = 0.074$; week 1-week 2: $\beta = 1.22$ [95% CI 0.22–2.22], adjusted $p = 0.077$), which was not observed for any other time interval. Microbiota stability in the first two weeks of life was also significantly

positively associated with IgG concentrations against all other pneumococcal vaccine serotypes (adjusted $p \le 0.083$, Supplementary Table 2). In contrast, no significant associations were found between microbiota stability and anti-MenC IgG concentrations.

The first two weeks of life, where gut microbiota stability was associated with anti-pneumococcal IgG concentrations, is compatible with the time frame when we previously found the largest difference in gut microbial composition between vaginally born and C-section born infants in this cohort (at the age of one week)[26]. In addition, this time frame coincides with the 'window of opportunity' when the gut microbiota primes the maturation of the immune system[16–18]. Therefore, we decided to focus on the microbial community composition in 'week one' samples, where we identified three distinct community state types (CSTs) (Supplementary Fig. 2). PERMANOVA confirmed that these CSTs differed considerably in community composition ($R^2 = 34.8\%$, $p < 0.001$). Infants with CST1 ($n = 55$) were characterized by a microbial community with low relative abundances of both *Bifidobacterium* and *E. coli*, while infants with CST2 ($n = 48$) had profiles with high relative abundances of *Bifidobacterium*, and infants with CST3 ($n = 16$) had high relative abundances of *E. coli* (Fig. 3a). Species-level microbial community composition obtained by shotgun sequencing of a subset of 20 'week one' samples confirmed that samples assigned to CST2 had high relative abundances of *Bifidobacterium breve* and/or *Bifidobacterium longum*, and samples assigned to CST3 had high relative abundances of *E. coli*, while samples assigned to CST1 mostly lacked these species (Supplementary Fig. 3).

We then studied whether these CSTs were associated with anti-Ps6B and anti-MenC IgG concentrations following vaccination. Infants with CST1 had the lowest IgG concentrations against both Ps6B and MenC (anti-Ps6B IgG: GMC 7.84 ng/ml [95% CI 4.88-12.60]; anti-MenC IgG: GMC 8.28 ng/ml [95% CI 5.93–11.56]; Fig. 3b). Compared with infants with CST1, anti-Ps6B IgG concentrations were approximately

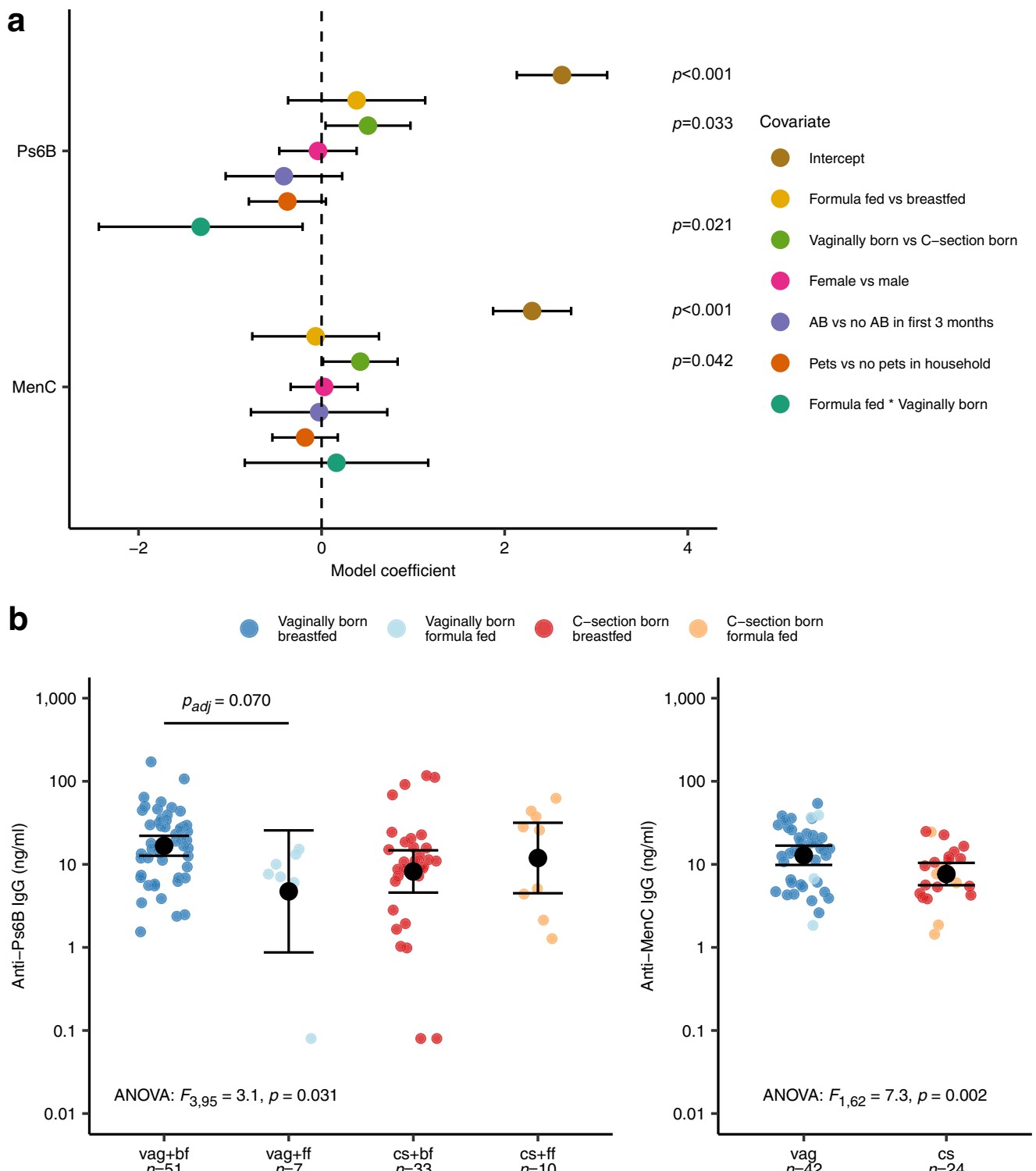

**Fig. 2 | Associations between early life exposures and anti-pneumococcal and anti-MenC IgG concentrations following vaccination. a** Data are presented as model coefficients with 95% CI per covariate computed with two-sided multi-variable linear regression with log-transformed anti-Ps6B (*n* = 101) or anti-MenC (*n* = 66) IgG concentrations as dependent variable. The analysis was not adjusted for multiple comparisons. C-section = cesarean section; AB = antibiotics. **b** anti-pneumococcal serotype 6B (anti-Ps6B) IgG responses for vaginally born, breastfed (vag + bf, *n* = 51), vaginally born, formula fed (vag + ff, *n* = 7), C-section born,

breastfed (cs + bf, *n* = 33), and C-section born, formula fed (cs + ff, *n* = 10) infants (left), and anti-meningococcus type C (anti-MenC) IgG responses for vaginally born (vag, *n* = 42) and C-section born (cs, *n* = 24) infants (right). Black dots and error bars represent geometric mean concentrations (GMCs) with 95% CI. Significance was assessed using two-sided analysis of variance (ANOVA) on log-transformed IgG concentrations followed by a post-hoc Tukey–Kramer test, correcting for time between vaccination and IgG measurements. $P_{adj}$ = FDR-adjusted *p*-value. Source data are provided in the Source Data file.

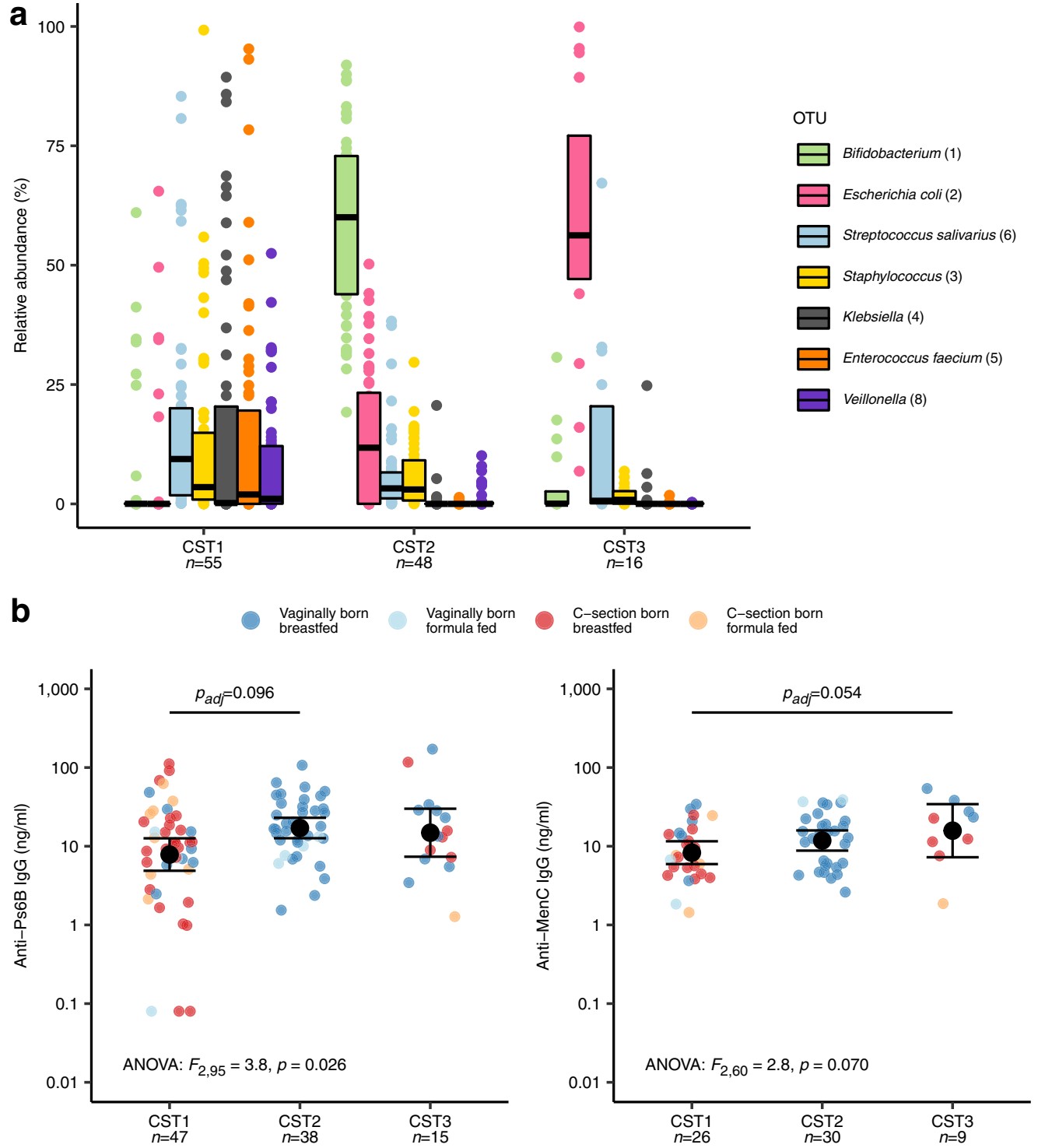

**Fig. 3 | Gut microbial community state types at week 1 and anti-Ps6B and anti-MenC IgG concentrations. a** Boxplot of relative abundances of the top 7 operational taxonomic units (OTUs) per community state type (CST) defined at 1 week of age. Boxes show medians with interquartile ranges. **b** CSTs are plotted against anti-Ps6B IgG concentrations (left) and anti-MenC IgG concentrations (right). Dots are colored according to mode of delivery and feeding type from birth. Black dots and error bars represent GMCs with 95% CI. Significance was assessed using two-sided ANOVA on log-transformed IgG concentrations followed by post-hoc Tukey–Kramer tests, correcting for time between vaccination and IgG measurements. $p_{adj}$ = FDR-adjusted $p$-value. Source data are provided in the Source Data file.

twofold higher in infants with CST2 (GMC 17.05 ng/ml [95% CI 12.64–23.00], adjusted $p$ = 0.096) as well as in infants with CST3 (GMC 14.85 ng/ml [95% CI 7.36–29.97], adjusted $p$ = 0.202), though only the comparison of anti-Ps6B responses between CST1 and CST2 infants was significant. We observed similar overall associations between week one CSTs and IgG responses against most other pneumococcal vaccine serotypes, but differences between CST1 and CST2 were not significant (Supplementary Table 3). By contrast, anti-MenC IgG concentrations in infants with CST3 were nearly twofold higher (GMC 15.76 ng/ml [95% CI 7.25–34.26], adjusted $p$ = 0.054) than in infants with CST1.

Mode of delivery was a strong driver of week one CSTs. All infants with CST2 were vaginally born, which was significantly more than

infants with CST1 (29.1%; Fisher's exact test, adjusted $p < 0.001$), or CST3 (62.5%, adjusted $p < 0.001$). Vaginal birth was also over-represented in infants with CST3 compared to CST1 (adjusted $p = 0.020$). In contrast, feeding type (breastfeeding vs. exclusive formula feeding) was not significantly different between these CSTs. A post-hoc analysis revealed that the association between mode of delivery and anti-Ps6B IgG responses disappeared with the addition of week one CST as an independent variable, indicating that the positive effect of vaginal delivery on anti-Ps6B IgG depended fully on the CST. In contrast, vaginal delivery remained significantly associated with anti-MenC IgG responses, regardless of week one CST, suggesting an independent effect (Supplementary Table 4).

To evaluate whether observed differences in early life microbial community composition were sustained for a prolonged time, including time points closer to vaccination, temporal development of the gut microbiota according to week one CST was assessed using PERMANOVA. The microbial community composition of children according to their CST defined at week one converged over time, resulting in no differences between samples belonging to the CST groups from month six onward (Fig. 4a). In pairwise comparisons, the observed differences in microbial community composition disappeared between infants with CST1 and CST3 by month one, between infants with CST2 and CST3 by month four, and between infants with CST1 and CST2 by month six. Similarly, relative abundances of *Bifidobacterium* and *E. coli* converged over time between CST groups (Fig. 4b). At the age of 12 months, we identified two distinct CSTs, which were not significantly associated with anti-Ps6B or anti-MenC IgG responses, confirming that early life microbiota were more strongly related to vaccine responses than the microbiota close to time of vaccination (Supplementary Fig. 4).

### Early life dynamics of individual OTUs were related to vaccine responses

Next, we investigated differences in individual OTU succession patterns within the first two months between high and low vaccine responders (stratified along the median antigen-specific IgG response). Higher abundances of *E. coli* (days 0–41, adjusted $p = 0.013$) and *Bifidobacterium* (days 0–5, adjusted $p = 0.027$) were associated with high anti-Ps6B responses (confirmed for 7/9 other pneumococcal vaccine serotypes, Supplementary Table 5). This was also observed for several *Bacteroides* OTUs, whereas *Clostridium, Prevotella* and *Streptococcus pyogenes* were associated with low responses (adjusted $p < 0.050$).

Higher *E. coli* abundance (days 0–13, adjusted $p = 0.072$) was also associated with high anti-MenC responses (Supplementary Table 6). Because the MenC vaccination is administered at the age of 14 months, which is much later in life than the pneumococcal vaccinations, we extended the analysis to 12 months to allow for identification of associations with OTUs that colonize later in life. In high anti-MenC responders, we observed significantly higher abundances of multiple low abundant OTUs belonging to the Lachnospiraceae family, including *Fusicatenibacter saccharivorans* (days 101–381, adjusted $p = 0.080$), *Pseudobutyrivibrio* (days 125–381, adjusted $p = 0.036$) and several *Blautia* and *Roseburia* OTUs (Supplementary Table 7).

### Species-specific validation using targeted qPCR

Finally, we performed a targeted species-specific qPCR to validate the presence and abundance of *E. coli, Klebsiella* spp. and *Enterococcus* spp. in all samples obtained at one week of age ($n = 119$). The relative abundance of *E. coli* showed a strong inverse correlation with *E. coli* Ct-values (Spearman's $\rho = -0.88$, $p < 0.001$), and the same was observed for *Klebsiella* spp. Ct-values (Spearman's $\rho = -0.41$, $p < 0.001$) and for *Enterococcus* spp. Ct-values (Spearman's $\rho = -0.88$, $p < 0.001$), corroborating our 16S rRNA gene sequencing-based data. In line with our findings, *E. coli* presence was more often detected by qPCR in infants who would subsequently have high anti-Ps6B IgG responses (34/49,

69%) than in infants with low anti-Ps6B IgG responses (25/50, 50%; $p = 0.078$). *E. coli* was also more often detected in week one samples of infants who were born by vaginal delivery (54/74, 73%) than in C-section born infants (20/44, 45%; $p < 0.001$). Presence of *Enterococcus* spp. or *Klebsiella* spp. were not associated with the anti-Ps6B IgG response. Also, none of the species identified by targeted qPCR were associated with the anti-MenC IgG response (Supplementary Table 8).

## Discussion

We studied interactions between early life exposures, gut microbial community development in the first year of life, and subsequent antibody responses in saliva against pneumococcal and meningococcal conjugate vaccination in a healthy birth cohort. A stable gut microbial community with high relative abundances of potentially beneficial bacteria in the first weeks of life, including *Bifidobacterium* and *E. coli*, was associated with high antibody responses to pneumococcal vaccination at 12 months of life. Furthermore, high *E. coli* abundance in early life was associated with high antibody responses to meningococcal vaccination at 18 months of life. Vaginal delivery was associated with high antibody responses to both vaccines, and, as we previously showed in this cohort[26], with the early life gut microbiota colonization patterns that we now associated with high antibody responses. Previous studies on associations between gut microbiota composition and serum antibody responses have focused on the microbiota near the time of vaccination[5,6,10,11,22]. However, our findings suggest that especially early life gut microbiota development may set the stage for robust immune responses to childhood vaccinations.

The period in which we identified associations between the gut microbiota composition and vaccine responses coincides with the critical window of opportunity spanning the first 100 days of life, when immune maturation is most affected by the early life gut microbiota[28]. In mice, the detrimental effects of antibiotic-induced gut microbiota disruption on host immunity, including vaccine responses, metabolism and even lifespan were shown to be particularly potent when exposure occurs in early life[21,29,30]. Relevant to the capacity to mount an effective antibody response to vaccination, the early life gut microbiota have been implicated in the shaping of the systemic B cell and immunoglobulin repertoire[19,20,31]. For instance, deficiency of IgA and IgG1 production in germ-free mice can be restored by microbial exposure[32]. In line, a culture-based study executed in human infants showed that the presence of *E. coli* and bifidobacteria in the gut in the first weeks of life was related to higher numbers of circulating CD27$^+$ memory B cells at four and 18 months of life[33]. In a recent microbiota-based study, lack of early bifidobacterial colonization was also linked to immune dysregulation at the age of three months, showing reduced levels of circulating plasmablasts, and naïve and transitional B cells[17]. This suggests that bacterial colonization patterns in early infancy drive B cell maturation, and have a lasting effect on, among others, adaptive immunity which may, for instance, be reflected in differences in antibody responses to infant and childhood vaccinations. In line with this concept, we found associations between gut microbiota community state types (CSTs) characterized by high relative abundances of *E. coli* and/or *Bifidobacterium* and low relative abundances of, among others, *Streptococcus, E. faecium* and *Klebsiella* in one-week-old infants, with higher antibody responses to vaccination months later in childhood.

Our study adds to an existing body of evidence for a positive effect of *E. coli* and *Bifidobacterium* on the immune response to vaccination. For instance, higher relative abundances of Gram-negatives including *E. coli* were associated with an adequate immune response against oral rotavirus vaccines[5]. Another study showed that treatment with the probiotic *E. coli* Nissle in a pig model enhanced the immune response to human rotavirus infection[34], providing a causal link. A potential mechanism whereby *E. coli* may influence vaccine responses was pinpointed by a study demonstrating that impaired antibody

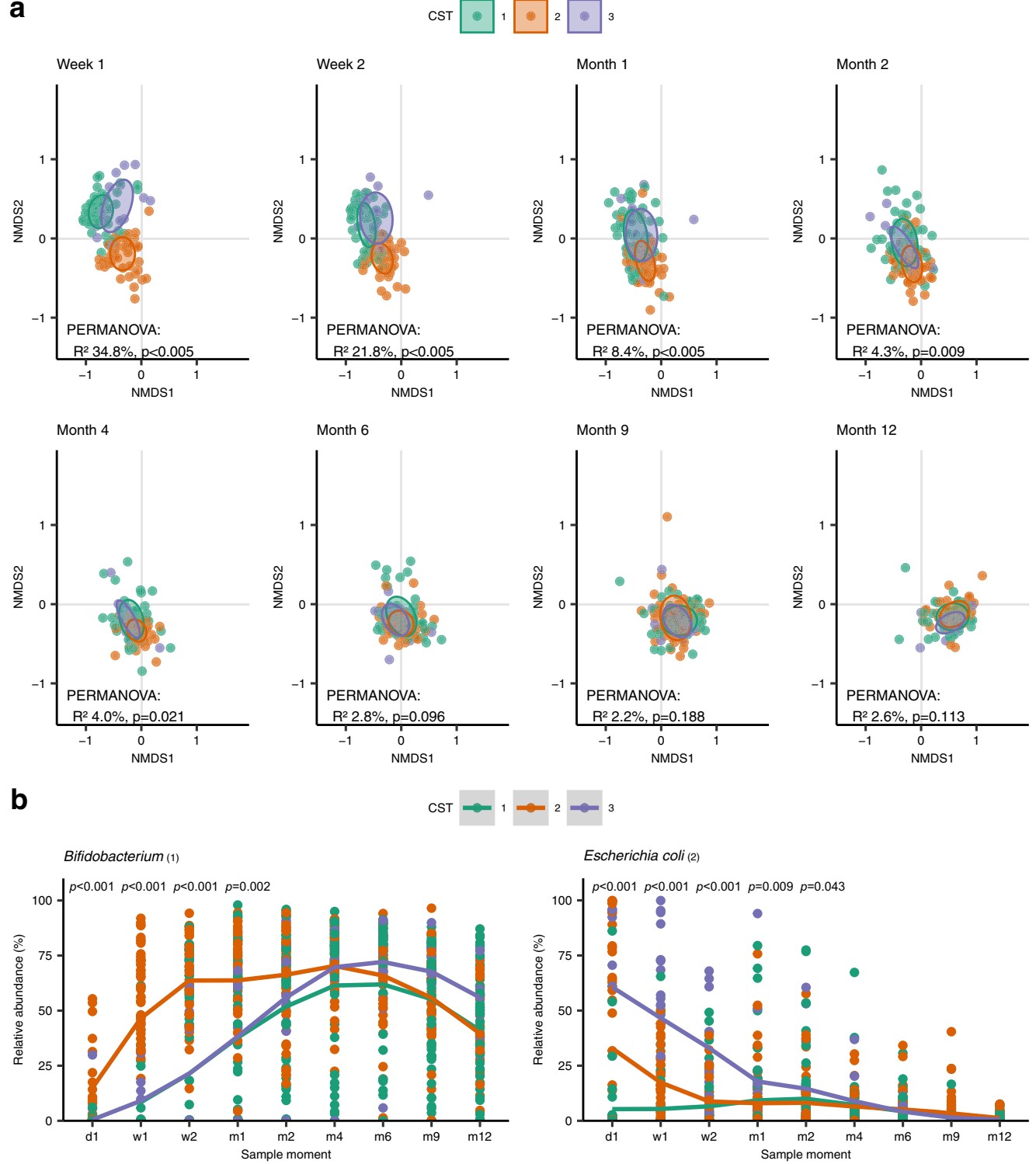

**Fig. 4 | Temporal gut microbial composition development according to week 1 CST. a** Non-metric multidimensional scaling (NMDS) plots based on Bray-Curtis dissimilarity, depicting the gut microbial composition per timepoint. Each dot represents the microbiota composition in a single participant's sample. Infants are stratified according to week 1 CST. Ellipses represent the standard deviation of data points for each CST. Effect sizes ($R^2$) calculated by permutational analysis of variance (PERMANOVA) and corresponding *p*-values are shown in the plots. **b** Relative abundances of *Bifidobacterium (1)* (left) and *Escherichia coli (2)* (right) over time according to week 1 CST. Significance of differences according to week 1 CST was assessed using Kruskal–Wallis tests. Source data are provided in the Source Data file.

responses to seasonal influenza vaccination in germ-free or antibiotic-treated mice were restored through TLR5-signaling by flagellated, but not unflagellated, *E. coli*[8], suggesting strain- and antigen-specific immune enhancement. Furthermore, early life absence of

*Bifidobacterium* has been associated with reduced systemic immune responses to *Bacillus* Calmette-Guérin, polio virus, tetanus and hepatitis B vaccination[11,22], which we also found for pneumococcal conjugate vaccination. *Bifidobacterium* species produce short chain fatty

acids (SCFAs) known to interact with host immune cells. For instance, early life reductions in fecal SCFAs have been linked to an increased risk of asthma[28], but effects of such metabolites on vaccine responses have not yet been studied. Conversely, we also found associations between other taxa such as *Clostridium*, *Prevotella* and *S. pyogenes* and lower vaccine responses, and it remains open to investigation whether these associations reflect a potential negative effect on the maturing immune system. Although the exact mechanisms remain to be unraveled, we hypothesize that very early life microbiota-host crosstalk at the intestinal mucosa imprints on systemic immunity, and may thereby affect vaccine responses.

Vaginal delivery and breastfeeding are important drivers of early life *Bifidobacterium* and *E. coli* abundance[13,26,35], whereas antibiotic treatment in the neonatal period has shown to dramatically reduce these bacteria[36]. Our results reveal an association between mode of delivery-induced early life microbiota profiles and anti-pneumococcal and anti-meningococcal vaccine responses, underlining that discouraging the increasing application of C-section in the absence of medical urgency may be important to preserve the microbiota-immune axis in infants. Whether antibiotic-induced microbiota disruption is associated with reduced vaccine responses has not yet been studied in infants[7,21]. Nonetheless, preterm infants, who generally receive antibiotic treatment in the first weeks after birth, have been shown to generate lower antibody levels following vaccination compared to term-born controls[37]. In our healthy, term-born cohort, very few infants were exposed to maternal antibiotics or required antibiotic treatment themselves in the first weeks of life, and further studies are required to compare our findings to (preterm) infants who received antibiotics as neonates.

We observed stronger associations of specific gut colonization patterns in early life with antibody responses to pneumococcal vaccination than with antibody responses to meningococcal vaccination. Furthermore, antibody responses against pneumococcal serotypes were not correlated to those against MenC, suggesting antigen-specific associations between the early life microbiota and vaccine responses. A more likely explanation is that pneumococcal and meningococcal vaccinations are administered at different ages. When meningococcal vaccination is administered at 14 months of age, the immune system has been exposed to other factors, and is already more mature and possibly more resilient to microbiota-related cues than when the first pneumococcal vaccination is administered at two months of age[16]. Notably, we associated higher abundances of members of the Lachnospiraceae family, including butyrate-producing taxa, with higher anti-meningococcal antibody responses. The abundance of these bacteria in the gut typically increases following the cessation of breastfeeding[35,38], and are generally found to be also beneficial for the developing immune system[39].

Perturbed gut microbial colonization patterns may contribute to reduced vaccine effectiveness across certain populations and settings[9]. Methods to modulate the gut microbiota following perturbations such as C-section birth are being investigated, and range from probiotic administration[40] to maternal fecal microbiota transplants[41], but it remains unknown if such interventions confer any long-term health benefits including enhanced vaccine immunogenicity. Our findings provide a rationale for investigations into potential interventions that modulate the infant gut microbiota to improve vaccine immunogenicity. Our results also suggest that different interventions should be considered for vaccinations given earlier in life compared to later in life in future studies.

Strengths of our work include the dense sampling at different timepoints, especially in the beginning of life. The extensively documented epidemiological data and microbiota composition of our cohort allowed us to establish associations between gut microbiota and vaccine responses in healthy infants. Furthermore, with the sensitive multiplex immunoassay technology, we could accurately measure antigen-specific antibody concentrations, even in very low volumes of saliva. Limitations of our work include using saliva for antibody measurements rather than serum for practical and ethical reasons. However, both anti-pneumococcal and anti-MenC vaccine-induced IgG concentrations in saliva were shown to correlate with serum concentrations[42,43], and are, therefore, a valid proxy for systemic IgG. Furthermore, while pneumococcal and meningococcal vaccination protect from infection primarily through neutralizing IgG, we did not assess other parameters of immunity such as IgA, antibody affinity, and T cell responses. Future studies could employ a multi-omics approach to obtain a complete overview of the mechanisms that underlie interindividual variation in vaccine responses[2,44]. Our observational study was also not primarily designed to study relationships between drivers, microbes and health outcomes such as antibody responses to vaccination, which limited our power to detect significant associations. Finally, the time between vaccination of the infants and sampling was variable and antibody measurement was not always performed within the optimal time window of 2–6 weeks after vaccination, which despite that we corrected for this in our analyses, may still have affected our results.

In conclusion, we demonstrate that mode of delivery-induced differences in the gut microbiota in the first weeks of life, including differences in *E. coli* and *Bifidobacterium* relative abundances, are associated with anti-pneumococcal and anti-MenC IgG responses to vaccination. Incorporating antibody responses to vaccination as a parameter in future trials of early life microbiota modulation could offer opportunities to assess beneficial outcomes on the microbe-mediated training of the immune system. Improved understanding of the microbial factors driving immune maturation and vaccine immunogenicity is key to improve vaccine performance and combat infectious diseases in children.

## Methods

### Study population and sample collection

Fecal samples, saliva and questionnaires were collected from a healthy birth cohort in which 120 healthy, full-term infants were enrolled. This study was primarily designed to investigate the effect of mode of delivery on early life microbiota development independent of intra-partum antibiotics, and therefore, routine peri-operative antibiotic administration to mothers delivering by C-section was postponed until after umbilical cord clamping. The current analysis of associations between host and microbial factors and antibody responses to vaccination entails a secondary goal of the study. Details on study design were previously published[26,45]. For the current analyses, we expanded our dataset with data and salivary samples up to 18 months from 78 (65%) subjects, who participated in the follow-up study beyond the first year of life. Both parents provided written informed consent. Ethical approval was granted by the Dutch national ethics committee (METC Noord-Holland, M012-015), and the study was registered in the Netherlands Trial Register under number NTR3986. Participants received no financial compensation. Study visits were conducted within 2 hours post-partum, 24–36 h after birth, at 7 and 14 days and at 1, 2, 4, 6, 9, 12 months and, for those who participated in the follow-up study, 18 months of age. Saliva for antibody measurement was collected at the ages of 12 and 18 months. An absorbent sponge (Malvern Medical Developments Ltd., Worcester, UK) was rubbed on the gums, cheek pouches and tongue for one minute. Saliva was immediately transferred to a tube containing EDTA (BD Vacutainer, New Jersey, USA) with protease inhibitor (Roche, Basel, Switzerland). Fecal samples for gut microbiota profiling were collected by the parents prior to each visit using sterile containers, and were directly stored in the home freezer, until collection by research personnel. Saliva and feces were transported on dry ice and stored at −80˚C awaiting subsequent laboratory analysis. Directly after birth, information on prenatal and perinatal characteristics was obtained. Glean Study Manager was used

to build a database for data collection (Sidekick-IT). At each subsequent home visit and additionally at the age of three months, extensive questionnaires including vaccination dates were collected. Infants received all routine childhood vaccinations from healthcare professionals at well-baby clinics according to the Dutch national immunization program (NIP), independent from the study. Ten-valent pneumococcal conjugate vaccine (PCV-10) was administered to infants born before September 2013 (52/120 participants) at the ages of 2, 3, 4, and 11 months, and to infants born from September 2013 (68/120 participants) at the ages of 2, 4, and 11 months due to changes in the NIP. Meningococcus group C (MenC) conjugate vaccination was administered at the age of 14 months.

## Measuring antibody responses to vaccination

Antigen-specific IgG against the capsular polysaccharides of pneumococcal vaccine serotypes 1, 4, 5, 6B, 7F, 9V, 14, 18C, 19F, and 23F was measured in saliva obtained at 12 months of age (approximately one month after the final PCV-10 dose), and IgG against MenC polysaccharide in saliva obtained at 18 months of age (approximately four months after vaccination). Antibodies were quantified using fluorescent bead-based multiplex immunoassays (MIA)[46–48]. Carboxylated microspheres (Luminex, Austin, TX) were coated with the respective polysaccharide antigens. To this end, antigens were first linked to Poly-L-lysine, and then the complex was bound to the microspheres in a reaction using EDC with sulpho-NHS. Standard reference sera with previously assigned concentrations of serotype-specific IgG were an in-house intravenous immunoglobulin (IVIG) for pneumococcal serotypes (Sanquin, Amsterdam, The Netherlands), calibrated on the WHO international standard 007sp (NIBSC), and CDC1992 for MenC (NIBSC, Ridge, United Kingdom)[49]. Saliva was thawed and centrifuged, and supernatants were diluted 1:2 and 1:10 using phosphate-buffered saline (PBS; pH = 7.2) with 5% antibody-depleted human serum (Valley Biomedical, Winchester, VA) and with 15 µg/ml multi cell wall polysaccharide (Statens Serum Insititut, Copenhagen, Denmark). From each dilution, 25 µl was mixed with an equal volume of beads. R-phycoerythrin conjugated goat anti-human IgG solution diluted 1:200 (Jackson ImmunoResearch, West Grove, PA) was added to each well. Analysis of the beads was performed on a BioPlex 200 apparatus using the BioPlex software package version 6.2 (Bio-Rad Laboratories, Hercules, CA). IgG concentrations were determined based on averaging results of both dilutions. When the concentrations differed more than twofold (coefficient of variation >47%), the result of the 1:10 dilution was used when in standard range. IgG concentrations were expressed in ng/ml. IgG concentrations below the lower limit of detection, which ranged from 0.08 ng/ml for pneumococcal serotype 4–0.37 ng/ml for pneumococcal serotype 14, and was 0.21 ng/ml for MenC, were set at half the lower limit of detection.

## DNA isolation and sequencing

For bacterial DNA extraction and microbiota profiling, fecal samples were first thawed and vortexed. Approximately 100 µl raw feces from each sample was added to 300 µl lysis buffer (Agowa Mag Mini DNA Isolation Kit, LGC ltd, UK), 500 µl 0.1-mm zirconium beads (BioSpec products, Bartlesville, OK, USA) and 500 µl phenol saturated with Tris-HCl (pH 8.0; Carl Roth, GMBH, Germany) in a 96-wells plate. The fecal samples were mechanically disrupted with a Mini-BeadBeater-96 (BioSpec products, Bartlesville, OK, USA) at 2100 oscillations per minute for 2 min. DNA purification was performed with the Agowa Mag Mini DNA Isolation Kit following the manufacturer's recommendations. Finally, the extracted DNA was eluted in 60 µl elution buffer (LGC Genomics, Germany). Adaptations in the standard DNA isolation procedure were applied for samples collected directly postpartum and on day 1, which were presumed to have low bacterial abundance and diversity[26]. The amount of bacterial DNA was determined by a quantitative polymerase chain reaction (qPCR), as described elsewhere[50],

using primers targeting the bacterial 16S rRNA gene (forward: CGAAAGCGTGGGGGAGCAAA; reverse: GTTCGTACTCCCCAGGCGG; probe: 6FAM-ATTAGATACCCTGGTAGTCCA-MGB) on the 7500 Fast Real Time system (Applied Biosystems, CA, USA). Samples with a minimum bacterial DNA yield of >0.3 ng/ul above the concentration in negative isolation controls were included in the sequencing protocol. The V4 hypervariable region of the 16S rRNA gene was amplified using F515/R806 primers (30 amplification cycles), and amplicon pools were sequenced on the Illumina MiSeq platform (Illumina, San Diego, CA) in 17 runs along with isolation and PCR blanks as negative controls.

## Bioinformatic processing

Sequences were processed in our bioinformatics pipeline[25]. We applied an adaptive, window-based trimming algorithm (Sickle, version 1.33) to filter out low quality reads below a Phred score threshold of 30 and/or a length threshold of 150 nucleotides[51]. Sequencing errors were corrected with BayesHammer (SPAdes genome assembler toolkit, version 3.5.0)[52]. Sets of paired-end sequence reads were assembled using PANDAseq (version 2.10) and demultiplexed (QIIME, version 1.9.1)[53,54]. Singletons and chimeras (UCHIME) were removed. Operational taxonomic unit (OTU) picking was performed with VSEARCH abundance-based greedy clustering of reads at 97% similarity[55]. Taxonomic annotation of OTUs was performed with the Naïve Bayesian RDP classifier (version 2.2) and the SILVA (version 119) reference database[56,57]. The resulting OTU table contained 6690 taxa. We selected OTUs that were present at a confident level of detection, i.e. representing at least 0.1% of all reads in at least two samples (excluding 0.4% of all reads)[12]. This abundance-filtered dataset contained 623 OTUs, and is referred to as the raw OTU table. We performed normalization by total sum scaling to obtain the relative abundance OTU table. Both OTU tables were used for downstream analyses.

## Whole genome sequencing for validation of OTU taxonomic annotations

Taxonomic annotations of the 16S rRNA gene sequences were validated, using whole genome shotgun sequencing (WGS) on a subset of 20 week one samples (ten from vaginally delivered infants, and ten from C-section born infants). For library preparation, the Truseq Nano gel free kit was used. From the libraries, 150 base paired-end sequence data were generated using a NovaSeq instrument to yield 750 M + 750 M reads in two runs. Reads were trimmed to remove amplicon adapter sequences and to maintain a quality threshold of 30 and a minimum read length of 35 base pairs using Cutadapt[58] (version 1.9.dev2). SAM files were generated per sample and per run with Bowtie2[59]. SAM files from different runs were merged per sample using Picard[60], and were used as input to MetaPhlAn2[61] for profiling and annotating the microbial communities within each sample (default parameters). The relative abundances of the top five 16 S rRNA gene sequencing-based OTUs *Bifidobacterium* (1), *E. coli* (2), *Staphylococcus* (3), *Klebsiella* (4), and *E. faecium* (5) were shown to correlate strongly with the WGS species-level relative abundances of *B. breve*, *B. longum*, and *B. adolescentis* (combined; Pearson's $\rho = 0.95$, adjusted $p < 0.001$), *E. coli* (Pearson's $\rho = 0.95$, adjusted $p < 0.001$), *Staphylococcus epidermidis* (Pearson's $\rho = 0.86$, adjusted $p < 0.001$), *Klebsiella oxytoca* (Pearson's $\rho = 0.83$, adjusted $p < 0.001$), and *E. faecium* (Pearson's $\rho = 0.92$, adjusted $p < 0.001$), respectively, confirming their taxonomies.

## Species-specific qPCR

Species-specific qPCR was performed on all week one samples ($n = 119$) to confirm the presence and abundance of *E. coli*, *Klebsiella* spp., and *Enterococcus* spp., using the VetMAX™ MastiType Multi Kit (Applied Biosystems™, CA, USA) according to the manufacturer's instructions. The qPCR test results were analyzed with the recommended Animal Health VeriVet Software, available on Thermo Fisher Cloud. One

sample was excluded from statistical analysis because its Internal Amplification Control did not pass the Ct-value criteria in three out of the four mixes.

## Statistics and reproducibility

Microbiome data were excluded from the analysis if fecal samples had insufficient bacterial DNA available ($n = 104$). Antibody measurements were excluded from the analyses if infants did not receive their vaccinations in time ($n = 8$ at month 12, $n = 1$ at month 18), or if the saliva sample did not have a sufficient volume for laboratory analysis ($n = 8$ at month 12, $n = 11$ at month 18; Supplementary Fig. 1). The study sample size was originally calculated to detect differences in the microbiota composition between infants born by vaginal delivery and by C-section[26]. For the current study, no statistical method was used to predetermine sample size. Data analysis was performed in R version 4.0.3 within RStudio version 1.3.1093[62]. All statistical tests were two-tailed, and p-values below 0.050 or Benjamini-Hochberg adjusted p-values below 0.100 were considered statistically significant. IgG responses were analyzed as continuous log-transformed variables or stratified along the median into high and low responses. All analyses were adjusted for time between vaccination and saliva collection using a second degree polynomial to account for the natural kinetics of the antibody response.

Concordance between IgG concentrations was evaluated using Pearson's correlations. Associations between early life host characteristics (mode of delivery, feeding type, sex, antibiotic use in the first three months, number of antibiotic courses, daycare attendance, having siblings and having pets) and IgG concentrations were assessed using univariate linear models, and factors with a $p < 0.050$ for one or more serotypes were included in multivariable models. IgG geometric mean concentrations (GMCs) were compared between groups defined by mode of delivery and feeding type, using ANOVA followed by post-hoc Tukey-Kramer tests to account for unequal group sizes (HSD.test-function, *agricolae* package [version 1.3-5][63], parameter 'unbalanced' set to TRUE). We tested the assumptions of normality and homogeneity of variance of the ANOVA test by inspecting the distribution of the residuals and with Levene's test, respectively.

Gut microbiota alpha diversity was assessed by the number of observed species and the Shannon diversity index (*phyloseq* package [version 1.38.0][64]). Associations between alpha diversity measures per timepoint and IgG concentrations were tested using linear models. Permutational multivariate analysis of variance (PERMANOVA) on the Bray-Curtis dissimilarity matrix was used to test for overall differences in the microbial community composition per timepoint between infants with high and low IgG responses (adonis2-function, *vegan* package [version 2.5-7][65]). Stability of the microbial community composition over time was calculated as the Bray-Curtis similarity (1–Bray–Curtis dissimilarity) between consecutive samples from the same individual, where a higher similarity indicates higher stability.

Dirichlet multinomial mixture models were used to group infants into community state types (CSTs) based on gut microbiota composition at week one and at month 12 separately (*DirichletMultinomial* package [version 1.36.0][66]). For this analysis, the raw OTU table was filtered, retaining OTUs present in >10% of the samples included in the analysis. The optimal number of CSTs was set at the number of Dirichlet components representing optimal model fit, testing a range of one to seven components. Model fit was based on the Laplace approximation to the negative log model, where a lower value indicates a better fit. Differences in the gut microbial community composition according to CST were evaluated using PERMANOVA (adonis-function, *vegan* package [version 2.5-7][65]). Differences in IgG GMCs according to week one and month 12 CSTs were evaluated using ANOVA and post hoc Tukey-Kramer tests, as described above.

Smoothing-spline analysis of variance (SS-ANOVA, fitTimeSeries-function, *metagenomeSeq* package [version 1.36.0][67,68]) was used to detect differences in individual OTU abundances in the first two months of life between infants with responses above and below the median antigen-specific IgG concentration. For the anti-MenC IgG response, this analysis was repeated for the entire 12 month follow-up period. For this analysis, the raw OTU table was filtered, retaining only OTUs present in >10% of all samples included in the analysis. This method detects differentially abundant OTUs, and identifies the time intervals in which significant differences exist.

Correlations between the relative abundances of *E. coli, E. faecium*, and *Klebsiella* at the age of one week and the species-specific Ct-values from targeted qPCR were evaluated with Spearman's rank-order correlations. $\chi^2$ tests were used to assess differences in presence of species identified by targeted qPCR between infants with above and below median IgG responses and between mode of delivery groups.

## Reporting summary

Further information on research design is available in the Nature Research Reporting Summary linked to this article.

## Data availability

Sequence data that support the findings of this study have been deposited in the NCBI Sequence Read Archive (SRA) database with BioProject ID PRJNA481243, and PRJNA555020. The vaccine response and relevant participant metadata are provided in the Source Data file. Additional participant metadata and data dictionaries can be made available after approval of a proposal. Taxonomic annotations were based on the Silva reference database (version 119). Source data are provided with this paper.

## Code availability

All R code used to run the statistical analysis is publicly available at https://gitlab.com/EMdK/muis_vaccine_responses.

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

## Acknowledgements

This work was funded by the Netherlands Organization for Scientific Research (NWO-VIDI; grant number 91715359, recipient: D.B.), Chief Scientist Office/NHS Research Scotland Scottish Senior Clinical Fellowship award (SCAF/16/03, recipient: D.B.), Spaarne Gasthuis, University Medical Center Utrecht, Dutch Ministry of Health, Welfare and Sport and the Strategic Program of the National Institute for Public Health and the Environment (SPR; grant number S/112009, recipient: S.F.). The authors are indebted to all the participating children and their families. We thank all the members of the research team of the Spaarne Gasthuis Academy, the laboratory staff, and the Streeklaboratorium Haarlem. We are grateful to Belinda van 't Land from Nutricia for providing some of the reagents.

## Author contributions

D.B., M.A.v.H., and E.A.M.S conceived and designed the study. M.A.v.H. was involved in enrolling the participants. M.L.J.N.C., F.v.d.H., and G.A.M.B. were responsible for the execution and quality control of the laboratory work. E.M.d.K., M.R., D.B., and S.F. analyzed the data. E.M.d.K., D.v.B., M.A.v.H., E.A.M.S., D.B., and S.F. wrote the paper. All authors significantly contributed to interpreting the results, critically revised the manuscript for important intellectual content, and approved the final manuscript. E.M.d.K., M.R., and M.L.J.N.C. have verified the microbiome and participant data. E.M.d.K., and F.v.d.H. have verified the antibody data.

## Competing interests

D.B. received funding from OM pharma and Sanofi. All authors declare no other competing interests.
