## [Peer Review File · Nature Communications]

Mode of delivery modulates the intestinal microbiota and
impacts the response to vaccinationREVIEWER COMMENTS

Reviewer #1 (Remarks to the Author):

This paper reports that infants born by natural delivery have higher IgG responses against meningococcal and pneumococcal vaccines compared to those delivered by Cesarean section, and that this is associated with different microbiota composition. The study is well done, the results are quite interesting and should be published. Unfortunately, the biological foundations of the observation are not investigated. The limitations of the study are well discussed on pages 15-16.

Reviewer #2 (Remarks to the Author):

This manuscript reported the association between mode of delivery, gut microbiota development and mucosal antigen-specific IgG responses against pneumococcal and meningococcal conjugate vaccination at ages 12 and 18 months in a prospective birth cohort of 120 infants. Results showed that natural delivery was associated with higher IgG responses against both vaccines, and the response was explained by higher abundances of Bifidobacterium and Escherichia coli in the first weeks of life.

Results are interesting and the paper is well-written and has clear objectives. However some points need to be addressed:

- How authors link the early microbiota (mainly Bifidobacterium and E.coli at 7 days) with IgG response at 12 and 18 months? Did the authors find differences according to mode of birth at 12 and 18m?? or within clusters? How the CST evolve during first 12-18m? Other factors would be related with the potential vaccine's response. In figure 2, authors showed the impact of different factors on the IgG antibodies, most of those factors are also related to changes in microbiota. Please expand the potential effect of confounding factors and the association of early microbiota and antibodies at 12-18m.
- At biological level, how specific bacteria present in 7days after birth would influence the impact of vaccines later in life (2, 4, 11 months). Please, include more information about potential mechanism. Furthermore, did other vaccines influence the response against pneumococcal and meningococcus vaccines? Did the repeated vaccination
- Did the authors confirm the link between Bifidobacterium with shotgun? what about species resolution? would some specific Bifidobacterium influence the antibodies levels? (Any info on probiotic consumption in the cohort?)
- Did authors check if the impact on antibodies response was mainly by mode of birth or antibiotic exposure?? did the authors include the number of antibiotic treatments during first 12 and 18m after birth as factor?
- How the microbiota stability was measured? would the differences observed in IgG antibodies reflect the differences in number of individuals included (fig.2B)?

Reviewer #3 (Remarks to the Author):

This study examines vaccine efficacy – antigen-specific IgG responses to two vaccines – as it relates to delivery mode and associated gut microbiota. A cohort of 120 healthy infants is used in the study. The introduction is relevant, concise, and well written and the discussion is well structured, focusing on relevant studies from the literature. Relationships between delivery mode, early-life gut microbiota, and vaccine response are identified, but much of the paper seems to be comparing gut microbiota at various time points post birth based on CST, which is correlated with birth mode. Birth mode effects on microbiota have been studied as have effects of the microbiota on vaccine efficacy and birth mode

on vaccine efficacy.

Major Comments

This study claims to examine efficacy for 2 vaccines in 120 infants, but in actuality, efficacy of 1 vaccine is measured in 120 infants and efficacy of 2 vaccines is measured in only a subset – 78 – which were followed through the period of the second vaccine's administration. Please update the abstract and first line of results to reflect this so as not to mislead the author.

In the statistics section, the relevant independent variables were lacking in some cases, making it difficult to understand what research questions the researchers are answering with each test. Also, what was the rationale behind the decision to compare CST using PERMANOVA in the same weeks that CST was classified? I would consider using PERMANOVAs to compare biologically relevant groups (above/below median IgG concentrations, delivery mode, etc) rather than groups already known to have distinct community state types. The smoothing spline analysis is well described and seems highly relevant.

Nonparametric tests were used to examine correlations between IgG concentrations, but parametric tests were used to compare between groups – please explain this decision. Was something done to correct for unbalanced design (eg. N = 51 vs 7; N = 26 vs 30 vs 9)?

Please expand the materials and methods section instead of sending the reader to previous publications. For example, why OTUs and not ASVs (97 vs 99%)? which pipeline was used for denoising? what were the QC cutoffs? PCR protocol?

In total the study is mainly descriptive, can the authors add experiments to test for the mechanism?

Minor Comments (by line)

78: I am happy to see qPCR was performed here, both for 16S and for specific taxa, but there is no further mention of this in the paper. Can you please expand the relevance of these tests? Using qPCR to validate abundance data is important and lacking in many studies. Having run qPCR for both the total community and specific taxa puts you in a prime position to validate results with true rather than relative abundances.

83: The filter of 0.1% relative abundance seems very strict – 90% of the data is lost. Can you please provide rationale for this strict filter? Are results consistent if a less stringent filter is used?

104: is this 10% of all samples or 10% of each group (above/below median)

120-121: here it says “ranged between” and then a single number is given. A range is in the (). Is the single number a mean? Median?

144: please updated here and elsewhere “naturally born” \diamond “vaginally born”

Table 1: Please add another column with summary statistics for the subgroup that continued to 18 months

Figure 4a: add significance to this figure or the legend

Figure 4b: can this be done with absolute abundance from your qPCR rather than the 16S data?

Reviewer #4 (Remarks to the Author):

NCOMMS-21-31736

In this manuscript, the authors evaluated correlative relationship between early gut microbiota, IgG response against pneumococcal and meningococcal vaccines and factors that influence microbiota. The major finding of this study was that high *E. coli* relative abundance was associated with higher anti-meningococcal and anti-pneumococcal IgG responses, and high *Bifidobacterium* relative abundance was associated with greater anti-meningococcal response.

The manuscript is well written; however, the conclusions and take-home messages are highly

speculative with the data being presented. Since the three clusters the authors observed in week one samples disappeared in later sampling points and the associations were performed based on those clusters, how do authors explain the relevance of those clusters to vaccination response which occurred much later? Why the authors did not consider taking samples during and after vaccination to match time points of saliva collection for IgG response? How stable are week one microbiota samples since week two samples did not show the same patterns? Additionally, the associations reported were based on relative abundance instead of actual abundance, hence, the results should be taken with precaution as the numbers could be simply because of over-representation of certain microbial phylotypes. Throughout the manuscript, the authors need to change abundance to relative abundance since the results do not reflect real quantities. A qPCR validation of Bifidobacterium and E. coli abundances would increase the strength of the findings. Another limitation is the use of V4 primers. This primer set is commonly used for human gut microbiota studies; however, it is not ideal for early life /infant microbiome studies since it does not capture as well as V1-V3 primers for the early life phylotypes. Lastly, how did the authors handle the multiple variables and their interference on each other's impact on the established associations?

Abstract: abstract is well written however please modify statements such as line 8 '.... Response could be explained' and line 11 '.... Childhood vaccines are mediated ...'. These statements are too speculative or too strong for the correlative associations presented in the study.

Introduction: well written

Line 34 – does those five common vaccines include the ones evaluated in this study?

Line 43 – Please explain why only IgG was monitored

Methods. A diagram showcasing the sampling times for different analysis would be helpful for the readers, especially different samples collected at different time points were correlated.

Line 60. Please specify how many and when fecal samples were collected.

Does sampling time for the saliva selected based on optimum immune response?

Results: Written clearly.

Line 223: OTUs associated with low response might be also important to investigate further.

Discussion: Well-written.

RESPONSE TO THE REVIEWER COMMENTS

We would like to thank the Editor and Reviewers for giving us the opportunity to revise our manuscript. We have addressed all their comments and updated the manuscript accordingly. We believe that this version is much improved, and therefore hope it to be suitable for publication.

Below we include a point by point response to reviewers (reviewers comments shown in *Italic*).

Reviewer #1 (Remarks to the Author):

This paper reports that infants born by natural delivery have higher IgG responses against meningococcal and pneumococcal vaccines compared to those delivered by Cesarean section, and that this is associated with different microbiota composition. The study is well done, the results are quite interesting and should be published. Unfortunately, the biological foundations of the observation are not investigated. The limitations of the study are well discussed on pages 15-16.

We thank the reviewer for the kind words about our work and its suitability for publication. We agree that we did not investigate the mechanisms behind the observed associations between the gut microbiota and vaccine responses, as this was beyond the scope of this (observational) study. As the reviewer points out, we have outlined these limitations in our manuscript.

However, we have now more clearly detailed our hypothesis on early-life microbiota priming of the immune system in the discussion (page 11-12, lines 212-251), including the following observations and references. Based on literature combined with our own data, we hypothesize that exposure to microbes like *Bifidobacterium* and *Escherichia coli* in the gut of healthy infants in the first weeks after birth is important for a proper development of the immune system. This concept, known as the ‘window of opportunity’, where the early life gut microbiota has the strongest effect on the maturation of the immune system¹, is reflected in differences in vaccine responses later in life. Several mouse studies have shown the effect of antibiotic-induced microbiota perturbations on host immunity and metabolism, which are most potent when the exposure occurs in early life²⁻⁴. In the discussion, we cited a number of studies that link early-life microbiota with the overall maturation of the immune system, including the development of B cells⁵⁻⁷.

The associations we found between the microbiota very early in life and vaccine responses at a later time point, are additional evidence of microbial priming on the immune system, and underline that this may have consequences for vaccine responses and potentially other diseases. Regarding the molecular mechanisms, much remains unknown, but mouse studies have for instance shown that *E. coli* signaling through the TLR5 receptor is essential for mounting an immune response against an influenza vaccine⁸. Bacterial metabolites like short-chain fatty acids (SCFA) may also play a role¹. SFCAs are known to interact with host immune cells and are produced by, among others, *Bifidobacterium*.

Reviewer #2 (Remarks to the Author):

*This manuscript reported the association between mode of delivery, gut microbiota development and mucosal antigen-specific IgG responses against pneumococcal and meningococcal conjugate vaccination at ages 12 and 18 months in a prospective birth cohort of 120 infants. Results showed that natural delivery was associated with higher IgG responses against both vaccines, and the response was explained by higher abundances of *Bifidobacterium* and *Escherichia coli* in the first weeks of life. Results are interesting and the paper is well-written and has clear objectives.*

We thank the reviewer for their interest and compliments on our work.

However some points need to be addressed:

*1. How authors link the early microbiota (mainly *Bifidobacterium* and *E.coli* at 7 days) with IgG*

response at 12 and 18 months?

We believe this comment aligns with the remarks from Reviewer #1, and therefore kindly refer to our answer above for a detailed explanation (also incorporated in our manuscript).

2. Did the authors find differences according to mode of birth at 12 and 18m??

or within clusters?

How the CST evolve during first 12-18m?

We have previously shown the effect of mode of delivery on the overall development and composition of the gut microbiota⁹. In this work, we showed that these differences persisted up to the age of two months in our cohort, but were no longer found between 2 and 12 months of age. We agree with the reviewer that this is important information for the reader, and therefore have added a summary of the most important findings of our previous work in the results section (page 6-7, line 98-102).

Unfortunately, we do not have data available on the gut microbiota composition at 18 months, but since mode of delivery-induced changes to the microbiota were no longer visible from month 2 up to one year of age, we do not expect any new associations following our window of investigation (i.e. 12 months).

Mode of birth was indeed also different between week 1 clusters. This is described on page 8-9, lines 147-150.

The development of the week 1 CSTs in the first year of life is described on page 9, lines 157-165.

Here we showed that with time, the composition of the different CSTs converged, reaching a similar composition by month six.

3. Other factors would be related with the potential vaccine's response. In figure 2, authors showed the impact of different factors on the IgG antibodies, most of those factors are also related to changes in microbiota. Please expand the potential effect of confounding factors and the association of early microbiota and antibodies at 12-18m.

We agree with the reviewer that other factors, including those in the analysis presented in figure 2 are also related to microbiota changes. We initially explored their associations with IgG antibodies in univariate analyses, to assess whether they could be potential confounding factors. We have now clarified in more detail that we took this step prior to the multivariable analysis shown in figure 2 (page 5, line 70-72). Multivariable analysis showed that only mode of delivery (vaginal delivery vs. caesarean section birth) was significantly associated with IgG antibodies against both the pneumococcal and the meningococcal vaccine. Furthermore, feeding type from birth (breastfeeding vs. exclusive formula feeding) showed a significant interaction with delivery mode and IgG responses, but only for the pneumococcal vaccine response. The other factors tested (i.e. pets, sex, antibiotics, etc.) were not significantly associated with IgG antibodies and, therefore, were not considered as confounding factors in downstream analyses.

Delivery mode and feeding type are the most important drivers of the early-life microbiota, and were also associated with vaccine responses in our cohort. In our study, we describe a chain of events where mode of delivery/feeding type affect the microbiota at a very early timepoint, which in turn was associated with vaccine responses later in life, which led us to consider mode of delivery and feeding type as early modulators, rather than as consistent confounding factors. We have now explained this more clearly in manuscript (page 6, lines 91-93).

4. At biological level, how specific bacteria present in 7days after birth would influence the impact of vaccines later in life (2, 4, 11 months). Please, include more information about potential mechanism.

We have included more information about different potential molecular mechanisms, such as the previously described mechanism by which *E. coli* may affect vaccine responses through TLR5 signaling. Alternatively, bacterial metabolites such as SCFAs have also been shown to affect the maturing of the immune system, but to our knowledge, this has not been investigated in the context of vaccine responses. We have expanded on the potential biological mechanisms on page 12, lines 232-246.

5. *Furthermore, did other vaccines influence the response against pneumococcal and meningococcus vaccines?*

All the infants included in the study received the same routine vaccinations according to the Dutch national immunization schedule. Therefore, if other vaccines influenced the vaccine response mounted against the pneumococcal and meningococcal vaccines, all infants would be similarly affected. We have now added this information in the text: “infants received all routine childhood vaccinations from healthcare professionals at well-baby clinics according to the Dutch national immunization program (NIP), independent from the study”, as this is indeed important information to the reader (page 16-17, line 339-341).

6. *Did the authors confirm the link between Bifidobacterium with shotgun? what about species resolution? would some specific Bifidobacterium influence the antibodies levels? (Any info on probiotic consumption in the cohort?)*

The abundance of the *Bifidobacterium (1)* OTU was indeed confirmed with shotgun sequencing on a subset of 20 fecal samples obtained at week 1. In our previous publication on this cohort, we described that there was a high correlation between the relative abundance of the *Bifidobacterium (1)* OTU and the combined relative abundance of *Bifidobacterium* species *breve*, *longum* and *adolescentis*⁹. We have now summarized these results in the Methods section as well (page 19, lines 397-408).

Due to the limited data available from shotgun sequencing (20 samples of week 1), we did not have sufficient power to detect direct associations at the species-level with vaccine responses. However, we included a bar plot of the species-level relative abundance of the 20 samples according to their CST allocation, and observed that the high *Bifidobacterium* abundance in CST2 could be explained by several species of the *Bifidobacterium breve* and *Bifidobacterium longum* (shown in Supplementary figure 3). CST2 was in turn associated with high anti-pneumococcal vaccine responses. We agree with the reviewer that these findings with species-level resolution are important information, especially for probiotics studies, and have therefore added a sentence describing this in our results section (page 8, lines 131-135).

We do not have data on the use of probiotics in our cohort.

7. *Did authors check if the impact on antibodies response was mainly by mode of birth or antibiotic exposure??*

did the authors include the number of antibiotic treatments during first 12 and 18m after birth as factor?

In this cohort, antibiotic administration during caesarean section birth was postponed until after clamping of the umbilical cord to limit direct antibiotic exposure to the infant. Caesarean section-born infants were thus not exposed to peri-operative antibiotics. We have added a sentence to the Methods section to inform the reader of this important aspect of our study design (page 16, lines 316-319).

Overall, the number of infants who were exposed to antibiotics during and following birth was therefore minimal (n=2, both related to maternal fever during delivery; this information was added to Table 1).

The number of antibiotic courses received in the first 12-18 months of life was also very low; most infants received no antibiotics during this time (we have added this information to Table 1). The number of antibiotic courses was not associated with the antibody responses to vaccination in

univariate analysis (information added to page 5, line 72). Antibiotic treatment in the first 3 months was significantly associated with a lower vaccine response against several pneumococcal serotypes in univariate analysis. Therefore, receiving antibiotics in the first 3 months of life, rather than the total number of antibiotic courses, is now also included in the multivariable analysis, where it showed not to be independently associated with the antibody responses against pneumococcal serotype 6B or meningococcus group C (figure 2).

8. How the microbiota stability was measured?

Microbiota stability was measured using the Bray-Curtis similarity (1-Bray-Curtis dissimilarity) distance between consecutive timepoints from the same individual, where higher similarity indicates higher stability. We now describe this more clearly in the Results (page 7, line 112-113) and Methods (page 20, line 438-440).

9. Would the differences observed in IgG antibodies reflect the differences in number of individuals included (fig.2B)?

We thank the reviewer for pointing this out to us. We have now revised the statistics supporting our stratified analysis of IgG concentrations according to mode of delivery and feeding type and have adapted figure 2B accordingly. Indeed, the analysis did not account for differences in group sizes, i.e. the unbalanced design. We changed the post-hoc test used after ANOVA to the Tukey-Kramer test, which can correct for unequal group sizes (*HSD.test*-function from the *agricolae* R-package, with parameter 'unbalanced' set to TRUE, now described in the Methods section on page 20, lines 428-430). While this change did not affect the conclusions of our manuscript, the previously observed differences in anti-pneumococcal serotype 6B IgG concentrations between the factor interactions vaginally born-breastfed infants and c-section born-breastfed infants was no longer significant (see figure 2B).

In addition, for the anti-meningococcal vaccine response we did not include comparisons according to feeding type as formula fed groups were very small (n=4 and n=6 for vaginally born and formula fed respectively), and instead only compared groups based on mode of delivery. In the multivariable model, feeding type was not associated with the anti-meningococcal vaccine response (figure 2A).

Reviewer #3 (Remarks to the Author):

This study examines vaccine efficacy – antigen-specific IgG responses to two vaccines – as it relates to delivery mode and associated gut microbiota. A cohort of 120 healthy infants is used in the study. The introduction is relevant, concise, and well written and the discussion is well structured, focusing on relevant studies from the literature. Relationships between delivery mode, early-life gut microbiota, and vaccine response are identified, but much of the paper seems to be comparing gut microbiota at various time points post birth based on CST, which is correlated with birth mode. Birth mode effects on microbiota have been studied as have effects of the microbiota on vaccine efficacy and birth mode on vaccine efficacy.

We thank the reviewer for their positive comments.

Major Comments

1. This study claims to examine efficacy for 2 vaccines in 120 infants, but in actuality, efficacy of 1 vaccine is measured in 120 infants and efficacy of 2 vaccines is measured in only a subset – 78 – which were followed through the period of the second vaccine's administration. Please update the abstract and first line of results to reflect this so as not to mislead the author.

We thank the reviewer for this observation. It is true that only a subset of our initial cohort of 120 infants was followed after one year of age until the 18-month timepoint, when meningococcal vaccine

response was measured. In addition, we removed some samples from the analysis because they had a low volume or because the infant had not received their meningococcal/pneumococcal vaccination prior to sample collection. To be exact, the antibody response to the pneumococcal vaccine was measured in 101 out of 118 infants with follow-up until 12 months, and the antibody response to the meningococcal vaccine was measured in 66 infants out of 78 infants with follow-up until 18 months. We have therefore adapted the abstract (page 2, lines 4-6) accordingly, as the reviewer suggested. We have also changed the flowchart (Supplementary figure 1) to provide all relevant information about the number of fecal and salivary samples that were collected and the number of samples that were included in our analysis. We still mention that the study initially enrolled 120 infants, because the gut microbiota data from all infants was used to e.g. determine the community state types. We indicate that salivary antibody responses were measured for a subset of infant on page 5, lines 53-57 and this now also becomes more clear from Table 1.

2. In the statistics section, the relevant independent variables were lacking in some cases, making it difficult to understand what research questions the researchers are answering with each test.

We thank the reviewer for this observation, and have now updated our statistics section for each analysis where appropriate (page 19-21, lines 416-462).

3. Also, what was the rationale behind the decision to compare CST using PERMANOVA in the same weeks that CST was classified? I would consider using PERMANOVAs to compare biologically relevant groups (above/below median IgG concentrations, delivery mode, etc) rather than groups already known to have distinct community state types.

The rationale for using PERMANOVA to compare CSTs at the week that they were classified (week one) was to confirm that the different CSTs indeed represented significantly different overall microbiota compositions. We changed the wording of the sentence describing this result to make this more clear to the reader (page 8, line 127-128). We thank the reviewer for the suggestion to use PERMANOVA on above versus below median IgG concentrations. We had indeed performed this analysis on all timepoints, but found no significant differences in the overall microbial composition. We have now included this observation in the Results section (page 7, line 109-112). We did not include the R^2 and p-values from this PERMANOVA in the manuscript, but we would be happy to provide them in the Supplementary material if the reviewer considers it pertinent. PERMANOVAs comparing vaginally born to c-section born infants were reported in a previous publication, and were therefore not repeated in this manuscript⁹. However, the main results from our previous publication are now briefly summarized in the Results section to clarify to the reader that the effect of mode of delivery on gut microbiota development has been studied before in this cohort (page 6-7, lines 98-102).

4. The smoothing spline analysis is well described and seems highly relevant.

We thank and agree with the reviewer that this is an important analysis.

5. Nonparametric tests were used to examine correlations between IgG concentrations, but parametric tests were used to compare between groups – please explain this decision. Was something done to correct for unbalanced design (eg. N = 51 vs 7; N = 26 vs 30 vs 9)?

We thank the reviewer for noticing this, which led us to revisit our statistical methods and make some adaptations:

We had indeed used non-parametric Spearman's rank order correlations to examine correlations between IgG concentrations against different serotypes. We used this method, because we applied it on the raw IgG concentrations, which have a highly skewed distribution. For the remainder of the analysis, the IgG concentrations were log-transformed. The distribution of the log-transformed IgG concentrations follows a bell-shaped curve, and therefore, we used parametric ANOVA's and linear

models. In addition, we tested the assumptions of normality and homogeneity of variance of the ANOVA test by inspecting the distribution of the residuals and with the Levene's test, respectively. We added these methods to the statistics section (page 20, line 430-432). If considered necessary we would be happy to include these analyses as supplementary material.

For consistency, we have now performed Pearson correlations (instead of Spearman's rank order correlations) on the log-transformed IgG concentrations, which yielded similar results (see updated figure 1B).

Regarding the unbalanced design, we have now changed the *post hoc* test used after ANOVA to the Tukey-Kramer test. This test can account for unequal group sizes. We used the *HSD.test*-function from the *agricolae* R-package, with parameter 'unbalanced' set to TRUE, now described in the Methods section on page 20, lines 428-430. This resulted in minor changes in values (but not of the overall findings), and we have adapted figures 2B and 3B and supplementary table 3 accordingly.

Additionally, as pointed out for Reviewer #2, question #9, for the anti-meningococcal vaccine response, we omitted the distinction according to feeding type because of the very small group sizes, and only compared groups based on mode of delivery.

6. Please expand the materials and methods section instead of sending the reader to previous publications. For example, why OTUs and not ASVs (97 vs 99%)? which pipeline was used for denoising? what were the QC cutoffs? PCR protocol?

We have expanded the Methods section (from page 16, line 313) according to reviewer's suggestions. It now includes more detailed information on the laboratory procedures, bioinformatic processing and statistical analysis.

Our custom bioinformatic pipeline which we have previously published¹⁰ and now summarized in the Methods section (page 18-19, lines 382-396), resulted in OTUs rather than ASVs. Moreover, we preferred to work with OTUs over ASVs to keep some balance between the number of features tested and the number of samples, given that the number of ASVs is generally much higher than the number of OTUs, and to be consistent with our previous publication using these data⁹. Also, we hope that with the addition of some confirmatory shotgun (page 19, lines 397-408) and species-specific qPCR data (page 10, lines 185-197), the reviewer agrees that the downside of less taxonomic resolution when using OTUs instead of ASVs is partially covered.

7. In total the study is mainly descriptive, can the authors add experiments to test for the mechanism?

We acknowledge that this is an observational study, therefore only descriptive. Studying the mechanisms would require additional models (mostly in animals) which is beyond the scope of the message of our manuscript. However, we refer to previous comments on our hypothesis on mechanisms to Reviewer #1, and Reviewer #2 question #4, which is also now included in more detail in the manuscript (page 11-12, lines 212-251).

8. Minor Comments (by line)

8.1 78: I am happy to see qPCR was performed here, both for 16S and for specific taxa, but there is no further mention of this in the paper. Can you please expand the relevance of these tests? Using qPCR to validate abundance data is important and lacking in many studies. Having run qPCR for both the total community and specific taxa puts you in a prime position to validate results with true rather than relative abundances.

We thank the reviewer for highlighting our efforts into the species level identification of relevant taxa. qPCR was performed on all week 1 samples to validate the taxonomy of *Escherichia coli*, *Enterococcus faecium* and *Klebsiella* OTUs. The Ct-values of these three species from qPCR showed

significant inverse correlations with the relative abundance of their corresponding OTUs, validating our 16S rRNA gene sequencing-based abundances. We have now added this result to the manuscript (page 10, lines 186-191).

Total 16S rRNA gene-based qPCR was done on all samples, which indeed gives us the opportunity to estimate abundances by multiplication of total bacterial density and relative abundance data and correcting for copy number variation, following the methodology previously used by other research groups^{11,12}. We used these absolute abundances to validate our result from figure 4B as suggested by the reviewer, which indeed resulted in similar findings (see the comment on figure 4B below). However, as the 16S rRNA qPCR method has similar biases as sequencing (i.e. primer coverage, inadequate DNA extraction, PCR inhibitors, etc.), and since results were similar, we decided to report the relative abundances for consistency.

Plotting the 16S rRNA gene qPCR data over time also shows that the bacterial density remained quite stable from week 1 onwards, comparable between all subsequent timepoints in our cohort (see figure below, if relevant we can also include it in supplementary material). Moreover, for the most important microbiota analyses in our study (defining community state types using Dirichlet multinomial mixture models¹³ and the smoothing spline analysis^{14,15}), we used the raw read counts and applied their own internal normalization methods, with default parameters. For clarity, we have added this information on these analyses to the statistics section (page 20, line 443 & page 21, line 455).

Figure. Total bacterial density as assessed using 16S rRNA gene qPCR in the first year of life.

8.2 83: The filter of 0.1% relative abundance seems very strict – 90% of the data is lost. Can you please provide rationale for this strict filter? Are results consistent if a less stringent filter is used?

We filtered OTUs present at a confident level of detection, i.e. 0.1% relative abundance in at least two samples in order to remove very rare taxa. While it leads to the removal of many low abundant OTUs, it discarded only a very small percentage of the reads (0.4%). We have now included the percentage of reads that were retained after the filtering step (99.6%) and an appropriate reference for the filtering method¹⁶ and rationale in the manuscript (page 18-19, lines 392-393).

8.3 104: is this 10% of all samples of 10% of each group (above/below median)

This concerns 10% of all samples included in the analysis, which we have clarified in the text (page 21, lines 455-456).

8.4 120-121: here it says “ranged between” and then a single number is given. A range is in the (). Is the single number a mean? Median?

We have corrected this sentence to: “Geometric mean concentrations (GMC) of IgG concentrations against the different pneumococcal vaccine serotypes ranged from 7.33 ng/ml (95% CI 5.75-9.33 ng/ml) for serotype 23F to 27.30 ng/ml (95% CI 22.14-33.67) for serotype 19F” (page 5, lines 57-59). We hope that these adaptations clarify that the numbers refer to the geometric mean concentrations per serotype, and that the indicated range is between serotypes with the highest and lowest geometric mean concentrations.

8.5 144: please updated here and elsewhere “naturally born” à “vaginally born”

We have changed all instances of “naturally born” to “vaginally born” throughout the text.

8.6 Table 1: Please add another column with summary statistics for the subgroup that continued to 18 months

We have added a column to Table 1 that summarizes participant characteristics for the subgroup who had anti-meningococcal IgG measured at the age of 18 months.

8.7 Figure 4a: add significance to this figure or the legend

We calculated effect sizes (R^2) and corresponding p-values using PERMANOVA per timepoint and we have now included these in figure 4A and explained them more clearly in the legend.

8.8 Figure 4b: can this be done with absolute abundance from your qPCR rather than the 16S data?

We thank the reviewer for this suggestion. As previously explained (question #8.1), we calculated absolute abundances by multiplying our total bacterial density data from the 16S qPCR with *Bifidobacterium* (1) and *Escherichia coli* (2) relative abundances and subsequently dividing by the median copy number (3 for *Bifidobacterium* and 7 for *E. coli*), as previously done by several other research groups^{11,12}. Absolute abundances calculated using this method have been shown to correlate well with species-specific qPCR data¹¹. Comparing the analysis shown in figure 4B with those using absolute abundances, we made the same observations as we did with relative abundances (see figure below). Because the results were very similar, and for consistency, we chose to show the relative abundance figure in the manuscript, but we would be happy to provide the figure below in the supplementary material.

Figure. Relative (a.) and absolute (b.) abundances of *Bifidobacterium* (1) and *Escherichia coli* (2) in the first year of life according to week 1 community state type (CST). Significance was assessed with Kruskal-Wallis tests and indicated by ***: $p < 0.001$; **: $p < 0.005$; *: $p < 0.050$.

Reviewer #4 (Remarks to the Author):

In this manuscript, the authors evaluated correlative relationship between early gut microbiota, IgG response against pneumococcal and meningococcal vaccines and factors that influence microbiota. The major finding of this study was that high E. coli relative abundance was associated with higher anti-meningococcal and anti-pneumococcal IgG responses, and high Bifidobacterium relative abundance was associated with greater anti-meningococcal response.

1. The manuscript is well written; however, the conclusions and take-home messages are highly speculative with the data being presented. Since the three clusters the authors observed in week one samples disappeared in later sampling points and the associations were performed based on those clusters, how do authors explain the relevance of those clusters to vaccination response which occurred much later?

We have adapted the discussion of the paper, so that it resonates better with the associative nature of our findings throughout. Regarding the relevance of early-life microbiota clusters and vaccine responses at later timepoints, we refer to our responses to questions from Reviewer #1 and Reviewer #2 question #4. We have expanded our hypothesis of potential mechanisms in the discussion (page 11-12, lines 212-251).

2. Why the authors did not consider taking samples during and after vaccination to match time points of saliva collection for IgG response?

This study was primarily designed to study effects of mode of delivery on microbiota development, and these results were previously published⁹. The evaluation of antibody responses to vaccination in saliva was a secondary goal of the study. As this was not designed as a vaccine study, vaccinations were administered by the Dutch healthcare facilities according to the national immunization schedule and not by the study staff. For practical reasons, saliva collection for antibody measurement was then

combined with sample collection for microbiota characterization. To account for bias that may have been introduced by the study design (e.g. variation in the age at which the infants received their vaccinations), we adjusted all of our analyses for differences in time between vaccination and sampling. We have highlighted these limitations in the text, including that the current work concerned a secondary analysis (page 16, lines 319-321), and that the vaccines were administered by healthcare facilities, independent from the study (page 16-17, lines 339-341).

3. How stable are week one microbiota samples since week two samples did not show the same patterns?

It is well known that the microbiota strongly changes over the first months of life, so we expected to see such differences. In this cohort, we observed strong developmental patterns with low overall community stability, especially in the first 2 months of life⁹. The goal of our study was to investigate a possible chain of events where environmentally driven microbiome development in the beginning of life was associated with vaccine responses later on, which supports the hypothesis of early-life immune modulation by gut microbes.

The reasons why we focused on ‘week one’ samples specifically, were both data-driven and context-driven:

1. We found initial differences in microbiota stability between infants with high and low vaccine responses around this timepoint;
2. We identified week 1 as the timepoint when mode of delivery-associated microbiota changes were the most pronounced in a previous publication⁹;
3. The week 1 time point fell within the critical ‘window of opportunity’ when immune maturation is most sensitive to bacterial cues from the gut microbiota, which is why we believe that bacterial succession in the earliest phase of life may be highly relevant for overall immune maturation and vaccine responses^{1,5}.

We have now highlighted our motivations more clearly in the results section (page 7, lines 121-125).

4. Additionally, the associations reported were based on relative abundance instead of actual abundance, hence, the results should be taken with precaution as the numbers could be simply because of over-representation of certain microbial phylotypes. Throughout the manuscript, the authors need to change abundance to relative abundance since the results do not reflect real quantities. A qPCR validation of Bifidobacterium and E. coli abundances would increase the strength of the findings.

We appreciate the comment that our results are based on relative abundances, with the associated limitations. We have adapted ‘abundances’ to ‘relative abundances’ throughout the text where appropriate. For example, this was not done for the part of the results section describing the differential abundance testing with smoothing spline analysis. This method uses an internal normalization procedure on raw read counts (cumulative sum scaling)^{14,15} and hence does not assess differences in relative abundance.

In our dataset, we performed a targeted qPCR validation of the relative abundance of *E. coli*, *Enterococcus* spp., and *Klebsiella* spp. on all week 1 samples. The Ct-values for these species showed significant inverse correlations with the relative abundances of their corresponding OTUs, and we have added this result to the manuscript (page 10, lines 186-191). In addition, as pointed out in question #8.8 of reviewer #3, we used our qPCR results of the total bacterial fraction to calculate absolute abundances of, among others, *Bifidobacterium* and *E. coli* (qPCR 16S rRNA gene concentration x relative abundance, corrected for copy number differences^{11,12}). This metric was previously shown to correlate very strongly with species-specific *Bifidobacterium* and *E. coli* qPCR results¹¹. We used these data to recreate figure 4B, and found that the result was highly comparable to our findings using relative abundances (see figure at question #8.8 from reviewer #3). For consistency and because qPCR also has its biases (as pointed out in response to question #8.1 from reviewer #3),

we chose to show the figures using Relative abundances in the manuscript, but we would be happy to provide this additional information in the supplementary material if preferred.

5. Another limitation is the use of V4 primers. This primer set is commonly used for human gut microbiota studies; however, it is not ideal for early life /infant microbiome studies since it does not capture as well as V1-V3 primers for the early life phylotypes.

We agree that a study has shown that, for early-life microbiome studies, the V1-V3 or V7-V9 regions perform better than V4-V5 primers, especially for *Bifidobacterium*¹⁷. Nonetheless, these results cannot be directly extrapolated to ours, as we used V4 primers and not V4-V5. The V4 region is still the most commonly targeted region in 16S rRNA gene sequencing-based microbiome studies, including early-life^{18,19}.

Furthermore, we used shotgun sequencing on a subset of 20 week 1 samples to validate our findings obtained with 16S rRNA gene sequencing, and found very strong correlations between the most abundant OTUs and the most abundant species obtained with shotgun sequencing. For *Bifidobacterium* specifically, the *Bifidobacterium* (1) OTU correlated very strongly with the top 3 *Bifidobacterium* species identified by whole genome sequencing (*B. breve*, *B. longum* and *B. adolescentis*). These findings reassured us that the V4 sequencing protocol had accurately captured the microbial community composition in our samples. Correlations between 16S rRNA gene sequencing and shotgun sequencing results were previously published⁹, and are now summarized in the methods section of our manuscript (page 19, lines 397-408).

6. Lastly, how did the authors handle the multiple variables and their interference on each other's impact on the established associations?

We first tested associations between host factors and the vaccine responses. We focused on early-life factors that are known to influence microbiota development and vaccine immunogenicity, which were mode of delivery (vaginal birth versus C-section), feeding type from birth (breastfeeding versus exclusive formula feeding), sex, having pets, having older siblings, antibiotic use, and daycare attendance. We first tested whether there were associations between these factors and anti-pneumococcal and anti-meningococcal IgG responses using univariate analysis. Factors with a significant result for at least one serotype were included in multivariable analysis, which were mode of delivery, feeding type, sex, having pets and antibiotic use in the first 3 months of life. This is now better explained in the methods section (page 20, lines 423-427).

We additionally included an interaction term between feeding type and mode of delivery, because these variables were related to each other (C-section born infants more often received exclusive formula feeding). The result of this analysis is presented in figure 2, which shows that only mode of delivery and feeding type were significantly associated with vaccine responses. These factors are also important drivers of microbiota development from birth²⁰.

We next established associations between the early-life microbiota and vaccine responses. We noted that the aspects of the microbiota we associated with vaccine responses (e.g. CSTs at week 1) were also associated with mode of delivery. Because this sequence of events where host factors present at/from birth influenced the microbiota only in early life, which in turn was associated with vaccine responses, we considered mode of delivery and feeding type as early modulators, rather than as confounding factors. We have now explained this more clearly in manuscript (page 6, lines 91-93). The statistical analysis section was improved to clarify which independent variables were included in which analysis.

7. Abstract: abstract is well written however please modify statements such as line 8 '... Response could be explained' and line 11 '... Childhood vaccines are mediated ...'. These statements are too speculative or too strong for the correlative associations presented in the study.

We have adapted these statements to better reflect the associative nature of our findings (page 2, lines 7-12). In line with these changes, we have also adapted the title of our manuscript.

8. Introduction: well written

Line 34 – does those five common vaccines include the ones evaluated in this study?

This study indeed investigated the immune response to the 13-valent pneumococcal conjugate vaccine and the meningococcus group C conjugate vaccine, among others. We have adapted the sentence so that it includes this information (page 3, lines 31-32).

9. Line 43 – Please explain why only IgG was monitored

We acknowledge that the immune response to vaccination entails more than IgG responses, such as T cell responses and other antibody isotypes. However, the humoral antibody response is considered the most important immune effector of the vaccines we studied (pneumococcal and meningococcal conjugate vaccines)²¹. We have added this information and reference in the introduction (page 4, lines 42-43). For both anti-pneumococcal and anti-meningococcal IgG, it has previously been shown that concentrations in saliva correlate well with concentrations in blood^{22,23}, and are vaccine-induced while IgA may also be boosted by natural pneumococcal colonization²⁴. Hence, IgG is the best proxy for systemic vaccine-induced humoral immunity in saliva, which we used instead of blood for practical and ethical reasons. We have expanded the information on this subject in the discussion (page 14, lines 290-294). That said, using IgG as the sole parameter to assess vaccine responses is mentioned as a limitation of our study (page 14, lines 294-296). We agree with the reviewer that it would be optimal to study vaccine responses in the broadest sense, preferably using a multi-omics approach, which we propose as an avenue for further research in our manuscript (page 14, line 296-298).

10. Methods. A diagram showcasing the sampling times for different analysis would be helpful for the readers, especially different samples collected at different time points were correlated.

Line 60. Please specify how many and when fecal samples were collected.

We have now adapted the flowchart (Supplementary figure 1) showing the number of fecal samples that were collected and that were included in the analysis per timepoint, and the number of saliva samples collected and included in the analysis at 12 months and at 18 months. The timepoints at which fecal samples were collected are now also more clearly specified on page 16, lines 327-329.

11. Does sampling time for the saliva selected based on optimum immune response?

The samples were obtained at the first sampling moment in our prospective cohort that followed the (final) vaccination. The optimal time window for antibody measurement following vaccination is generally considered to be within 2-6 weeks. The anti-pneumococcal vaccine response sampling timepoint was at the age of 12 months, which was a median of 28 days after vaccination and within the ideal sampling window. However, for the anti-meningococcal vaccine response, there was a median of 116 days between vaccination (at age 14 months) and sampling (at age 18 months), which is a later than optimal. To account for this, all our statistical analyses were adjusted for the time between vaccination and sampling. We have included this as a limitation (page 14, line 300-303), and included in the methods section that this was a secondary analysis on a prospective cohort study (page 16, lines 319-321).

12. Results: Written clearly.

Line 223: OTUs associated with low response might be also important to investigate further.

We agree with the reviewer that this is also an important question for future research. We have added this in the manuscript (page 12, lines 246-248).

Discussion: Well-written.

References

1. Arrieta, M. C. *et al.* Early infancy microbial and metabolic alterations affect risk of childhood asthma. *Science Translational Medicine* **7**, (2015).
2. Cox, L. M. *et al.* Altering the intestinal microbiota during a critical developmental window has lasting metabolic consequences. *Cell* **158**, 705–721 (2014).
3. Lynn, M. A. *et al.* Early-Life Antibiotic-Driven Dysbiosis Leads to Dysregulated Vaccine Immune Responses in Mice. *Cell Host and Microbe* **23**, 653–660.e5 (2018).
4. Lynn, M. A. *et al.* The composition of the gut microbiota following early-life antibiotic exposure affects host health and longevity in later life. *Cell Reports* **36**, 109564 (2021).
5. Henrick, B. M. *et al.* Bifidobacteria-mediated immune system imprinting early in life. *Cell* **184**, 1–15 (2021).
6. Li, H. *et al.* Mucosal or systemic microbiota exposures shape the B cell repertoire. *Nature* **584**, (2020).
7. Zeng, M. Y. *et al.* Gut Microbiota-Induced Immunoglobulin G Controls Systemic Infection by Symbiotic Bacteria and Pathogens. *Immunity* **44**, 647–658 (2016).
8. Oh, J. Z. *et al.* TLR5-mediated sensing of gut microbiota is necessary for antibody responses to seasonal influenza vaccination. *Immunity* **41**, 478–492 (2014).
9. Reyman, M. *et al.* Impact of delivery mode-associated gut microbiota dynamics on health in the first year of life. *Nature Communications* **10**, 4997 (2019).
10. Bosch, A. A. T. M. *et al.* Maturation of the Infant Respiratory Microbiota, Environmental Drivers, and Health Consequences. A Prospective Cohort Study. *American Journal of Respiratory and Critical Care Medicine* **196**, 1582–1590 (2017).
11. Jian, C., Luukkonen, P., Yki-Järvinen, H., Salonen, A. & Korpela, K. Quantitative PCR provides a simple and accessible method for quantitative microbiome profiling. *PLOS ONE* **15**, e0227285 (2020).
12. Lloréns-Rico, V., Vieira-Silva, S., Gonçalves, P. J., Falony, G. & Raes, J. Benchmarking microbiome transformations favors experimental quantitative approaches to address compositionality and sampling depth biases. *Nature Communications* **12**, (2021).
13. Holmes, I., Harris, K. & Quince, C. Dirichlet Multinomial Mixtures: Generative Models for Microbial Metagenomics. *PLoS ONE* **7**, 30126 (2012).
14. Paulson, J. N., Talukder, H. & Bravo, H. C. Longitudinal differential abundance analysis of microbial marker-gene surveys using smoothing splines. *bioRxiv* (2017) doi:10.1101/099457.
15. Paulson, J. N., Stine, O. C., Bravo, H. C. & Pop, M. Differential abundance analysis for microbial marker-gene surveys. *Nature Methods* **10**, 1200–1202 (2013).
16. Subramanian, S. *et al.* Persistent gut microbiota immaturity in malnourished Bangladeshi children. *Nature* **510**, 417–421 (2014).
17. Alcon-Giner, C. *et al.* Optimisation of 16S rRNA gut microbiota profiling of extremely low birth weight infants. *BMC Genomics* **18**, 1–15 (2017).
18. Stewart, C. J. *et al.* Temporal development of the gut microbiome in early childhood from the TEDDY study. *Nature* **562**, 583–588 (2018).

19. Stokholm, J. *et al.* Delivery mode and gut microbial changes correlate with an increased risk of childhood asthma. *Science Translational Medicine* **12**, eaax9929 (2020).
20. Bokulich, N. A. *et al.* Antibiotics, birth mode, and diet shape microbiome maturation during early life. *Science translational medicine* **8**, 343ra82 (2016).
21. Siegrist, C.-A. Vaccine Immunology. in *Plotkin's Vaccines* (eds. Plotkin, S. E., Orenstein, W. A., Offit, P. A. & Edwards, K. M.) 16-34.e7 (Elsevier, 2018).
22. Rodenburg, G. D. *et al.* Salivary Immune Responses to the 7-Valent Pneumococcal Conjugate Vaccine in the First 2 Years of Life. *PLoS ONE* **7**, 1–8 (2012).
23. Stoof, S. P. *et al.* Salivary antibody levels in adolescents in response to a meningococcal serogroup C conjugate booster vaccination nine years after priming: Systemically induced local immunity and saliva as potential surveillance tool. *Vaccine* **33**, 3933–3939 (2015).
24. Bogaert, D. *et al.* Pneumococcal conjugate vaccination does not induce a persisting mucosal IgA response in children with recurrent acute otitis media. *Vaccine* **23**, 2607–2613 (2005).

REVIEWER COMMENTS

Reviewer #1 (Remarks to the Author):

The authors did a good job in addressing the reviewers' comments. I am ok with this version and I don't have additional comments.

Reviewer #2 (Remarks to the Author):

Authors have covered all my questions and concerns.
The paper content and clarity have been improved.
No more comments

Reviewer #3 (Remarks to the Author):

I thank the authors for their answers. However, I still think the study is descriptive and suffers from a lack of mechanistic insight.